# Policy Optimization under Imperfect Human Interactions with Agent-Gated Shared Autonomy

**Zhenghai Xue**[1]    **Bo An**[1,2]    **Shuicheng Yan**[2,3]
[1]Nanyang Technological University, Singapore    [2]Skywork AI
[3]National University of Singapore

## Abstract

We introduce AGSA, an **A**gent-**G**ated **S**hared **A**utonomy framework that learns from high-level human feedback to tackle the challenges of reward-free training, safe exploration, and imperfect low-level human control. Recent human-in-the-loop learning methods enable human participants to intervene a learning agent's control and provide online demonstrations. Nonetheless, these methods rely heavily on perfect human interactions, including accurate human-monitored intervention decisions and near-optimal human demonstrations. AGSA employs a dedicated gating agent to determine when to switch control, thereby reducing the need of constant human monitoring. To obtain a precise and foreseeable gating agent, AGSA trains a long-term gating value function from human evaluative feedback on the gating agent's intervention requests and preference feedback on pairs of human intervention trajectories. Instead of relying on potentially suboptimal human demonstrations, the learning agent is trained using control-switching signals from the gating agent. We provide theoretical insights on performance bounds that respectively describe the ability of the two agents. Experiments are conducted with both simulated and real human participants at different skill levels in challenging continuous control environments. Comparative results highlight that AGSA achieves significant improvements over previous human-in-the-loop learning methods in terms of training safety, policy performance, and user-friendliness. Project webpage is at https://agsa4rl.github.io/.

## 1 Introduction

Human-in-the-loop Learning (HL) methods (Kelly et al., 2019; Celemin et al., 2022) integrate human participants in the training process of RL and facilitate safe-guarded RL training without relying on environment rewards. Existing HL methods leverage low-level human involvement in two main aspects: (1) Monitoring the agent training process for potential safety violations (Peng et al., 2021; Luo et al., 2024) and intervening agent control when necessary; (2) Providing online demonstrations during intervention (Li et al., 2022b; Peng et al., 2023). However, human participants may exhibit suboptimal behaviors (Xue et al., 2023c;a) when either monitoring or providing demonstrations. For example, human participants can be unfamiliar with the task requirements or the interface for shared autonomy. They may get tired as training goes on and fail to figure out whether the learning agent is in a dangerous situation. Network latency may also occur stochastically when human participants perform remote operations (Mandlekar et al., 2020). When interacting with embodied agents, human participants may struggle to control all joints and carry out a high-level policy instead (Li et al., 2022a). Therefore, one critical challenge of HL is *how to improve training safety and efficiency in face of unpredictable imperfections of human interactions?*.

To address the challenge of imperfect human monitoring, recent methods have shifted from human-gated training to agent-gated approaches, where a separate gating agent oversees the environment interaction of the learning agent and calls for human intervention when necessary. EnsembleDAgger (Menda et al., 2019) uses high uncertainties in decision making as the trigger of human intervention. But as we demonstrate in Sec. 3.1, dangerous regions may have low uncertainty after they are visited for a few times, so such heuristic criteria often fail to detect dangerous regions and safeguard the learning agent. To handle imperfect human demonstrations, some approaches attempt to model human behavior through environment reward and request intervention only when the learning agent is

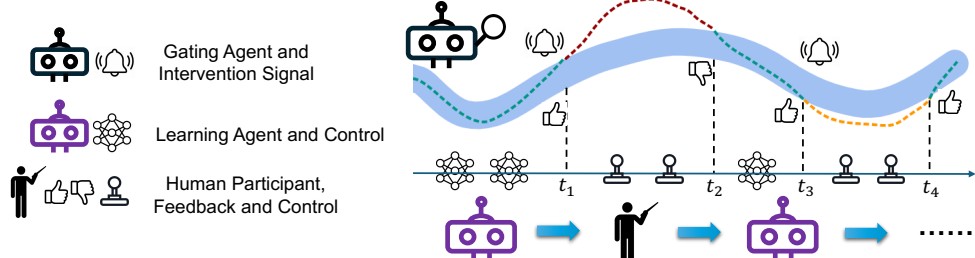

Figure 1: The learning agent (in purple) interacts with the environment under the monitoring of the gating agent (in black). The gating agent decides when to request human intervention. Learning agent trajectories are in green and human trajectories are in red and yellow. Human feedbacks are denoted with thumbs up and down. Feedbacks at $t_1$ and $t_3$ are human evaluations on whether the gating agent triggers control switch at proper timesteps. Feedbacks at $t_2$ and $t_4$ are human preferences on whether the current intervention trajectory is better than the previous one. For example, the trajectory between $t_3$ and $t_4$ is better than that between $t_1$ and $t_2$, so human may provide positive feedback on $t_4$.

likely to act incorrectly (Xue et al., 2023d; Liu et al., 2023b). But they assume access to environment rewards which are not available in reward-free settings.

In this work, we propose a novel **A**gent-**G**ated **S**hared **A**utonomy (AGSA) framework that simultaneously addresses both aspects of imperfect human interactions, as shown in Fig. 1. AGSA is built upon the agent-gated training pipeline and learns from human feedback, both on whether the gating agent proposes intervention at proper timesteps and on whether the current human intervention trajectory is better than the previous one. Conceptually, rather than fully relying on low-level human monitoring or human demonstrations—both of which can be imperfect—we assume the accuracy of high-level human feedback, as it is easier for humans to make *relative* judgements that compare trajectories as better or worse, than to provide *absolute* optimal decisions (Helson, 1964; Kahneman & Tversky, 2013). The reliance on human feedback empowers recent success of applying RL from Human Feedback (RLHF) to train large language models (Ouyang et al., 2022), but has not been thoroughly investigated in HL for continuous control tasks. As in RLHF, we train reward models that capture human preferences, which are further used to train gating value functions that estimate the long-term effect of the gating actions. In this way, the gating agent can provide more accurate, human-aligned, and forseeable intervention signals than previous methods. To train the learning agent without environment reward, we regard states that require intervention as undesirable, assigning negative proxy rewards to state-action pairs that precede human intervention. Since the gating agent fully controls human interventions, the learning agent is insulated from imperfect human demonstrations.

Theoretical analyses show that optimizing human feedback provides performance and safety bounds for the mixed behavior policy that interacts with the environment, demonstrating the effectiveness of the gating agent. Meanwhile, training with negative proxy rewards ensures a lower-bound performance guarantee for the learning agent. For empirical evaluations, we select two challenging continuous control tasks of robotic locomotion and autonomous driving, using the MuJoCo (Todorov et al., 2012) and MetaDrive (Li et al., 2023) simulator. We employ neural policies with varying performance levels, along with human participants inexperienced in evaluation tasks, to provide imperfect human involvement. Comparative results demonstrate that AGSA learns efficiently from imperfect data while maintaining overall training safety. Our contributions in this paper can be summarized as follows: (1) We identify the challenges posed by imperfect low-level human control and propose to utilize high-level human feedback instead. (2) We design a novel framework for agent-gated shared autonomy, where the gating agent is trained with human feedback and the learning agent is trained with intervention decisions from the gating agent. (3) We provide both theoretical and empirical evidence to support the efficiency and safety of the proposed framework.

## 2 BACKGROUND

### 2.1 PRELIMINARIES

To model agent-gated shared autonomy, we consider two Markov Decision Processes (MDPs) for the learning agent and the gating agent. The learning MDP is defined by the tuple $M_l = \langle \mathcal{S}, \mathcal{A}_l, T_l, \gamma, d_0 \rangle$

including a state space $\mathcal{S}$, an action space $\mathcal{A}_l$, a transition function $T_l$, a discount factor $\gamma$, and an initial state distribution $d_0$. The gating MDP is defined by the tuple $M_g = \langle \mathcal{S}, \mathcal{A}_g, T_g, \gamma, r_g, d_0 \rangle$ with the same state space $S$, discount factor $\gamma$, and initial state distribution with $M_l$. $A_g = \{0, 1\}$ is the binary indicator of whether to let human policy $\pi_h$ intervene. $r_g$ is the learned reward function for training the gating agent. Policies of the gating MDP $\pi_g(s)$ have deterministic binary outputs and are regarded as gating functions, i.e., $\pi_g(s) = 1$ denotes human intervention and control, and $\pi_g(s) = 0$ denotes learning agent's control. The overall behavior policy, or the data collection policy, can be defined as $\pi_b(\cdot|s) = (1 - \pi_g(s))\pi_l(\cdot|s) + \pi_g(s)\pi_h(\cdot|s)$, where $\pi_l$ is the policy in the learning MDP. The goal of agent-gated shared autonomy is to optimize the learning policy $\pi_l$ and maximize its expected return $\eta(\pi_l) = \mathbb{E}_{\tau \sim d_0, \pi_l, T_l}[\sum_0^\infty \gamma^t r(s_t, a_t)]$, where $r$ is the inaccessible environment reward function. Therefore, there is no form of human involvement during testing. The state-action value function for $\pi_l$ is defined as $Q_l(s, a) = \mathbb{E}_{\pi_l, T}[\sum_{t=0}^\infty \gamma^t r(s_t, a_t)|s_0 = s, a_0 = a]$. $Q_g$ for $\pi_g$ is defined as $Q_g(s, a) = \mathbb{E}_{\pi_b, T}[\sum_{t=0}^\infty \gamma^t r_g(s_t, a_t)|s_0 = s, a_0 = a]$.

## 2.2 RELATED WORK

**Human-Gated Shared Autonomy**  Human-gated HL algorithms rely on human participants to monitor the environment interaction of the learning agent and intervene on dangerous or repetitive states. HG-DAgger (Kelly et al., 2019) and IWR (Mandlekar et al., 2020) let human participants provide corrective demonstrations after intervention and perform Imitation Learning (IL) on human sampled trajectories. CEILING (Chisari et al., 2022) takes evaluative feedback with human demonstrations, assigning different weights in the imitation loss. Other algorithms combine HL with RL under human monitoring. Under the reward-free setting, RL agents resort to human-gated intervention for proxy feedback. HACO (Li et al., 2022b) and PVP (Peng et al., 2023) train the Q-value function by maximizing it on $(s, a)$ pairs from human generated trajectory and minimizing it on $(s, a)$ pairs from the agent. RLIF (Luo et al., 2024) assigns a reward of -1 to state action pairs that are one-step prior to human-gated intervention. While some human-gated shared autonomy methods, such as CEILING and RLIF, take suboptimal human demonstrations into consideration, they can still be negatively influenced by inaccurate human monitoring. Instead, this paper introduces a separate gating agent and no longer relies on humans to monitor the training process.

**Agent-Gated Shared Autonomy**  Existing agent-gated methods include EnsembleDAgger (Menda et al., 2019) which estimates uncertainty in decision making and asks for human intervention when the uncertainty level is high. But uncertainty is only an empirical criterion and cannot be aligned with human instructions. Liu et al. (2023a) introduce model-based failure prediction that foresees potential danger in a few steps. But the prediction still learns from human interventions that can be inaccurate. EGPO (Peng et al., 2021) lets human intervene if the learning agent has low action likelihood under human's policy distribution. ThriftyDAgger (Hoque et al., 2021) and BCVA (Gokmen et al., 2023) use goal reaching rewards to learn proxy value functions. Human intervention will be triggered if the proxy value drops below pre-defined thresholds. TS2C (Xue et al., 2023d) and AdapMen (Liu et al., 2023b) compare the value functions of the agent action and human action, and only let human intervene if their actions are guaranteed to have better outcomes. But human policy distribution or environment reward are hardly accessible in many real-world applications. The reliance on such information hinders broader applications of these methods. We request additional human feedback to train the gating agent. The feedback is collected when the human intervention starts and terminates, which is easy to implement and adds minimal burden to human participants.

We leave relevant researches on reward-free RL in Appendix A, where we mainly discuss the advantage of our framework over Preference-based RL (PbRL) methods.

## 3 POLICY OPTIMIZATION WITH AGENT-GATED SHARED AUTONOMY

In this section, we first provide motivating examples in Sec. 3.1 and discuss the drawbacks of previous agent-gated methods. Then we discuss our approach of training a long-term gating value function from human feedback in Sec. 3.2. In Sec. 3.3, the learning agent is trained from proxy reward signals based on the intervention decisions of the gating agent. We conduct theoretical analysis on the presented training framework in Sec. 3.4 and conclude the section with practical algorithm pipeline in Sec. 3.5.

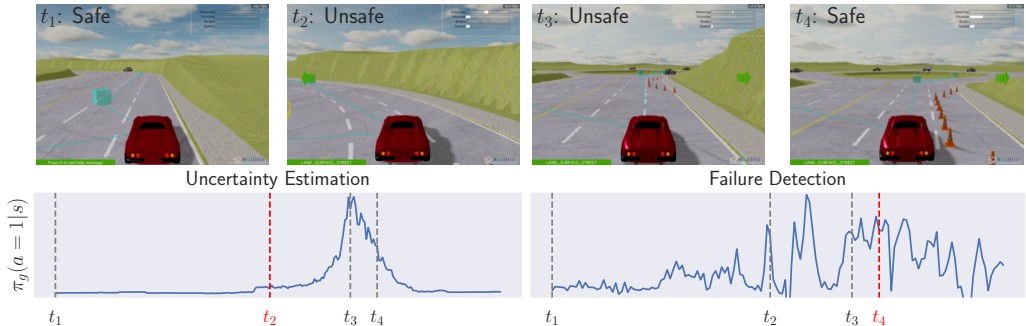

Figure 2: Probabilities of the gating agent requesting human intervention, with the uncertainty estimation method and the failure detection method. Timesteps highlighted in red have problematic intervention probabilities, either failing to recognize danger or being overly conservative.

### 3.1 MOTIVATING EXAMPLE

Among existing agent-gated shared autonomy algorithms, the uncertainty estimation method (Menda et al., 2019) triggers human intervention when state-action uncertainty exceeds a pre-defined threshold. The failure detection method (Liu et al., 2023a) imitates human intervention decisions. In Fig. 2, we illustrate the probabilities of both methods requesting human intervention along a trajectory in the MetaDrive (Li et al., 2023) simulator. In the presented trajectory, $t_1$ and $t_4$ are safe steps which should exhibit low intervention tendencies, but the failure detection method assigns a high intervention probability at $t_4$. This is likely due to human participants being overly conservative in the presence of nearby dangerous zones, leading to unnecessary intervention even when the learning agent operates correctly. $t_2$ and $t_3$ are dangerous steps due to incorrect vehicle direction, but the uncertainty estimation method assigns a low intervention probability at $t_2$. This is because state-action uncertainty is related to the complexity of environment components that is poorly aligned with the actual dangerous zones.

These examples highlight the limitations of current agent-gated algorithms, which cannot ensure appropriate timing to switch to human control. Instead of learning from potentially inaccurate human intervention decisions, in AGSA the gating agent first makes intervention decisions itself and then learns from human evaluative feedback on whether the intervention decisions are appropriate. AGSA also learns from human preference feedback on subsequent trajectories influenced by intervention decisions, getting rid of the heuristic criterion in the uncertainty estimation method.

### 3.2 TRAINING GATING AGENT FROM HUMAN FEEDBACK

The motivating examples demonstrate that to train an effective gating agent, relying solely on heuristics or step-wise imitation of human instructions is insufficient, mainly because these metrics are loosely connected to the training process of the learning agent. Overall, the central role of the gating agent is to help train the learning agent safely and effectively. Such a role can be characterized by the performance of the mixed behavior policy $\pi_b$, as it is in charge of collecting meaningful training data in the environment and avoid safety violations. Therefore, we analyse the impact of policy switching on the long-term performance of the mixed behavior policy $\pi_b$. We employ the gating value function $Q_g$ to quantify such long-term effect. $Q_g$ takes environment states $s$ and the binary intervention decisions $a_g$ as input. The gating policy can derived from $Q_g$ by selecting the gating action with higher long-term value:

$$\pi_g(s) = \begin{cases} 1 & \text{if } Q_g(s,1) > Q_g(s,0), \quad \text{(Human Intervention)} \\ 0 & \text{otherwise.} \quad\quad\quad\quad\quad\quad\quad \text{(Agent Control)} \end{cases} \tag{1}$$

To properly train the gating value function $Q_g$ for optimal intervention timing, human participants follow three steps, as illustrated in Fig. 3 (upper):

1. Providing a binary signal $I(s_t)$ that assesses whether the current environment state is indeed worth intervention. Such human evaluation provides a direct feedback on whether gating agent's intervention decisions successfully indicates dangerous or unexplored areas. Its advantage over directly imitating human intervention decisions is that humans have time to examine the intervention quality, rather than making real-time decisions that can be influenced by tiredness, carelessness, or network latency.

Figure 3: The framework for training both the gating agent and the learning agent. **Upper:** The gating agent generates actions $a_t^g \in \{0, 1\}$ to determine whether human intervene is required, receiving rewards $r_t^g$ based on human feedback. **Lower:** The learning agent generates actions $a^\pi$ to interact with the environment. Its rewards $r^\pi$ are set to -1 on states preceding human intervention and to 0 on other states.

2. Interacting with the environment for $T$ steps and offering online demonstration segment $\sigma = (s_t, a_t^{\text{human}}, \ldots, s_{t+T-1}, a_{t+T-1}^{\text{human}})$, aiming at guiding the learning agent out of the region that is dangerous or no longer needs exploration.

3. Providing a preference signal $p_t = P_\psi[\sigma \succ \sigma'] \in \{0, 0.5, 1\}$, indicating whether current segment $\sigma$ is better than the previous segment $\sigma'$. As human participants are familiar with recent trajectories of themselves, this way of online preference pair construction saves the burden for humans of reviewing previously sampled trajectories, as shown by the user study in Sec. 4.2. Bad human samples can happen due to imperfect human behaviors or untimely intervention decision of the gating agent. By assigning low human preference on these samples, the gating agent can learn from human demonstrations at all performance levels and mistakes in the intervention decision made by itself.

As RL environments usually have high-frequency actions, we allow humans to continuously intervene for $T$ steps to provide more accurate preference feedback, where $T$ is a predefined hyperparameter. The preference reward model $r_\psi$ is trained from human preference signals with the Bradley-Terry model (Bradley & Terry, 1952; Lee et al., 2021):

$$P_\psi[\sigma \succ \sigma'] = \frac{\exp \sum_{(s,a) \in \sigma} r_\psi(s, a)}{\exp \sum_{(s,a) \in \sigma} r_\psi(s, a) + \exp \sum_{(s',a') \in \sigma'} r_\psi(s', a')}, \tag{2}$$

$$\mathcal{L}^{\text{Reward}} = -\mathbb{E}_{(\sigma, \sigma', p_t)} \left[ (1 - p_t) \log P_\psi[\sigma' \succ \sigma] + p_t \log P_\psi[\sigma \succ \sigma'] \right]. \tag{3}$$

The overall reward function $r_g$ to train the gating value function $Q_g$ is the linear combination of the evaluation feedback $I(s_t)$ and the preference reward: $r_g(s_t, a_t^g) = I(s_t) + \lambda \sum_{n=0}^{N-1} r_\psi(s_{t+n}, a_{t+n}^{\pi_b})$, where $\lambda$ is the hyperparameter for reward balancing and $a_t^{\pi_b}$ denotes actions from the learning agent or human, depending on the control switching decision. The gating value function can therefore be trained with standard value-based RL methods with $r_g$. Besides being able to measure long-term performance, this training procedure does not require human to monitor or provide feedback during the learning agent's control. Compared to human-gated methods that require constant human oversight, this approach achieves more efficient utilization of human involvements and is more user-friendly, as demonstrated by the human study in Sec. 4.2.

### 3.3 TRAINING LEARNING AGENT FROM INTERVENTION SIGNALS

When training the learning agent in the reward-free setting, direct imitation will lead to degraded policy performance due to potentially suboptimal human demonstrations. So we need to design a proxy reward model $r_\pi$ on the training data. One straightforward approach is to use the learned reward model $r_\psi$ as $r_\pi$. But as shown in Tab. 2, the learning agent cannot benefit much from $r_\psi$, mainly because of the instability of $r_\psi$ that keeps updating. Instead, we propose to set $r_\pi$ based on the binary actions of the gating agent $a_t^g$, which are generated through comparisons of gating values and filter out most of the noisy signals. As shown in Fig. 3 (lower), state-action pairs that precede control switching, such as $(s_2, a_2^\pi)$ in Fig. 3, are likely to result in suboptimal outcomes and are assigned with a negative reward $r_\pi(s_2, a_2^\pi) = -1$. Other agent-generated state-action pairs, such as $(s_1, a_1^\pi)$ and $(s_5, a_5^\pi)$, receive a zero reward $r_\pi = 0$. Therefore, $r_\pi$ can be set as follows:

$$r_\pi(s_t, a_t^\pi) = \begin{cases} -1 & \text{if } a_{t+1}^g = 1, \\ 0 & \text{otherwise.} \end{cases} \tag{4}$$

While the learning agent may have erroneous actions far before the intervention, these mistakes cannot be identified without accurate human monitoring and access to environment reward. We instead rely on the ability of RL to perform implicit credit assignment (Pignatelli et al., 2024), allowing the agent to correct mistaken actions. This credit assignment problem is less challenging than that in tasks with sparse rewards (Rengarajan et al., 2022) thanks to the relatively dense intervention signals.

Human-generated samples, such as $(s_3, a_3^{\text{human}})$ and $(s_4, a_4^{\text{human}})$ in Fig. 3, are expected to guide the agent out of the dangerous or stagnant regions. However, given the potential suboptimality of human demonstrations, the learning agent is not directly trained on human-generated samples and proxy rewards $r_\pi$ are undefined for these samples. From the perspective of the learning agent, the trajectory temporarily terminates at $s_t$ when $t + 1$ is the human intervention step. Agent control will resume at $s_{t+T}$ ($s_5$ in Fig. 3) if the human successfully navigates through dangerous or unexplored areas. Otherwise, the environment will be reset to the initial state. In this way, the learning agent benefits from human-guided state recovery while remaining unaffected by imperfect human demonstrations.

### 3.4 THEORETICAL ANALYSIS

We provide theoretical justifications for the proposed training framework of AGSA. One important evaluation criteria of the gating agent is the ability of the mixed behavior policy $\pi_b(\cdot|s) = (1 - \pi_g(s))\pi_l(\cdot|s) + \pi_g(s)\pi_h(\cdot|s)(*)$, which is used to interact with the environment and collect training samples[1]. A well-performing $\pi_b$ facilitates efficient exploration and safety protection for the learning agent. In the following theorem, we show that the gating agent of AGSA can secure a performance lower bound of the behavior policy.

**Theorem 3.1.** *With the gating policy $\pi_g$ defined in Eq. (1) and $Q_g$ trained with $r_\psi$, the behavior policy $\pi_b$ defined in Eq. (*) has the following performance lower-bound[2]: $\eta(\pi_b) \geqslant \max\{\eta(\pi_h), \eta(\pi_l)\} - \frac{2\varepsilon_r}{(1-\gamma)^2}$, where $\varepsilon_r = \max_{s,a}|r(s,a) - r_\psi(s,a)|$ is the error of preference-based reward modelling.*

The bound contains the higher performance among the human and learning policy. This demonstrates that the gating agent facilitates efficient exploration by leveraging human demonstrations. Meanwhile, when human policies are suboptimal, the performance of the behavior policy will be lower-bounded by the learning policy itself, which deals with the issue of imperfect human demonstrations. In safety-critical scenarios, the step-wise training cost $c(s, a)$, i.e., the penalty on the safety violation during training, can be regarded as a negative reward. The gating agent can also provide safety guarantee for $\pi_b$, as discussed in Appendix B.2. We further show in the following theorem that the learning policy has a lower-bound performance guarantee when optimized with the proxy reward function $r_\pi$, demonstrating the effectiveness of the proxy reward from the intervention signal.

**Theorem 3.2.** *Let $\tilde{\pi}$ be the optimal policy trained with proxy rewards $r_\pi(s,a)$. $\tilde{\pi}$ has the following performance lower bound: $\eta(\tilde{\pi}) \geqslant \eta(\pi_h) - \frac{4\varepsilon_r}{(1-\gamma)^2}$.*

Similar performance lower-bounds are derived in previous human-in-the-loop methods with human-gated training (Luo et al., 2024) or with access to environment rewards (Liu et al., 2023b). AGSA obtains such lower bound with a milder assumption on the bounded error of preference-based reward modelling. Meanwhile, thanks to the gating agent that measures long-term intervention outcome, AGSA does not have a performance upper bound and may outperform imperfect human participants, as shown by the results in Sec. 4.

### 3.5 PRACTICAL ALGORITHM

Summarizing previous analysis, we present the detailed workflow of AGSA in Alg. 1. Line 5 and Line 10 construct the replay buffer $D_l$ for training the learning agent, assigning rewards based on gating agent outputs. Line 6 corresponds to three kinds of human interactions, including human demonstrations, human evaluative feedback on the intervention decision, and human preference feedback on the demonstrations. Line 7 constructs the replay buffer $D_g$ for training the gating value $Q_g$ and the dataset $D_p$ for training the reward model $r_\psi$. Line 8 denotes that the preference pair

---

[1]While human generated samples are mostly excluded when training the learning agent, the state $s_{t+T}$ at intervention termination will influence the subsequent agent-generated samples.

[2]Proofs to the theorems are in Appendix B.1.

---

**Algorithm 1** The practical workflow of AGSA.

---

1: **Input:** Gating value function $Q_g$; Learning agent policy $\pi_l$; Human policy $\pi_h$; Human preference model $P_\psi$; Reward model $r_\psi$; Learning agent replay buffer $D_l$; Preference Replay buffer $D_p$; Gating agent replay buffer $D_g$; Preference reward ratio $\lambda$; Human intervention steps $T$.
2: **for** epoch $i = 0, 1, 2, \ldots$ **do**
3:   **for** timestep $t = 1, 2, \ldots$ **do**
4:     **if** $Q_g(s_t, 1) > Q_g(s_t, 0)$ and not previous_intervene **then**
5:       Append $(s_{t-1}, a_{t-1}, s_t, -1)$ to $D_l$.
6:       Apply human policy $\pi_h$ for $T$ steps, getting trajectory segment $\sigma$; Query human for intervention evaluation $I(s_t)$ and preference feedback $p_t = P_\psi(\sigma \succ \sigma')$.
7:       Append $(s_t, 1, s_{t+1}, I(s_t) + \lambda \sum_{n=0}^{T} r_\psi(s_{t+n}, a_{t+n}))$ to $D_g$. Append $(\sigma, \sigma', p_t)$ to $D_p$.
8:       Set $\sigma' = \sigma$, previous_intervene=True, $t = t + T - 1$.
9:     **else**
10:       Append $(s_{t-1}, a_{t-1}, s_t, 0)$ to $D_l$ and $(s_t, 0, s_{t+1}, \lambda r_\psi(s_t, a_t))$ to $D_g$.
11:       Apply learning agent policy $\pi_l$ for 1 step; Set previous_intervene=False.
12:     Train $\pi_l, r_\psi, Q_g$ on $D, D_p, D_g$, respectively.

---

Table 1: Results of experiments with different performance levels of neural policies. Numbers are normalized scores according to D4RL (Fu et al., 2020). Numbers after $\pm$ are standard deviations across trials with four different seeds.

| Domain | Expert Level | DAgger | Ensemble-DAgger | Failure Prediction | BCVA | RLIF | AGSA (Ours) |
|---|---|---|---|---|---|---|---|
| Hopper | Low | 19.54 ± 2.14 | 11.91 ± 6.43 | 33.27 ± 7.36 | -29.69 ± 0.59 | 89.39 ± 9.76 | **94.18 ± 3.54** |
| | Medium | 38.70 ± 3.70 | 10.30 ± 6.68 | 50.02 ± 10.85 | -17.95 ± 21.13 | 92.40 ± 2.82 | **92.44 ± 3.83** |
| | High | 70.58 ± 9.74 | 39.44 ± 1.04 | 65.28 ± 12.32 | 55.27 ± 18.56 | 94.71 ± 1.04 | **95.79 ± 0.90** |
| | Average | 42.94 ± 5.19 | 20.55 ± 4.72 | 49.52 ± 10.18 | 2.55 ± 13.43 | 92.16 ± 4.54 | **94.14 ± 2.76** |
| Walker2d | Low | 12.37 ± 2.96 | -9.15 ± 3.30 | 23.50 ± 5.34 | -19.11 ± 9.63 | **115.24 ± 12.09** | 114.16 ± 2.17 |
| | Medium | 20.49 ± 3.15 | 23.93 ± 12.33 | 31.29 ± 8.07 | -14.53 ± 10.56 | 69.50 ± 38.44 | **109.38 ± 2.29** |
| | High | 57.94 ± 8.69 | 51.57 ± 1.22 | 50.82 ± 3.28 | 7.85 ± 45.88 | 65.53 ± 35.63 | **129.09 ± 2.98** |
| | Average | 30.27 ± 4.93 | 22.12 ± 5.62 | 35.20 ± 5.56 | -8.60 ± 22.02 | 83.43 ± 28.72 | **117.55 ± 2.48** |
| HalfCheetah | Low | 18.19 ± 1.97 | 11.42 ± 8.89 | 11.53 ± 1.29 | 47.45 ± 5.44 | 20.54 ± 2.93 | **83.01 ± 0.80** |
| | Medium | 31.53 ± 2.32 | 24.18 ± 0.35 | 15.91 ± 5.44 | 60.62 ± 7.40 | 15.79 ± 2.38 | **88.63 ± 0.20** |
| | High | 52.67 ± 5.77 | 28.99 ± 0.67 | 25.05 ± 4.67 | 71.99 ± 1.48 | 12.16 ± 3.62 | **89.66 ± 0.69** |
| | Average | 34.13 ± 3.35 | 21.53 ± 3.30 | 17.50 ± 3.80 | 60.02 ± 4.77 | 16.16 ± 2.98 | **87.10 ± 0.56** |

$(\sigma, \sigma')$ is constructed with the current and the previous human generated trajectory. In Line 13, $\pi_l$ and $Q_g$ can be trained with any value-based RL algorithms, such as TD3 (Fujimoto et al., 2018) and SAC (Haarnoja et al., 2018), and $r_\psi$ is trained with Eq. 3.

# 4 EXPERIMENTS

In this section, we conduct experiments to investigate the following questions: (1) Can AGSA facilitate efficient training and safe exploration in various challenging tasks, compared to previous human-in-the-loop training methods? (2) How does the components of AGSA, such as the evaluative and preference feedback to train the gating agent, contribute to its overall performance? (3) How do human participants evaluate AGSA in terms of performance alignment and interacting workload, compared with other algorithms? To answer these questions, we consider the task of robotic locomotion and autonomous driving, as shown by Fig. 5 in Appendix C.1. We conduct comparative analysis and ablation studies, as well as designing questionnaires for human-centered studies.

Experiments that involve real human participants are usually expensive and cost-sensitive. Their interaction can also exhibit large variance in different trials. Therefore, existing literature highly depend on trained neural policies as proxies for human policies (Peng et al., 2021; Xue et al., 2023d; Luo et al., 2024). We follow this setting in the robotics simulator MuJoCo (Todorov et al., 2012) and use neural policies with different performance levels to simulate imperfect human policies. We also conduct experiments with real human participants in the autonomous driving simulator MetaDrive (Li et al., 2023). Though neural policies or human participate are involved in the training process, all reported metrics in this section are obtained by the learning agent alone in separate evaluation rollouts.

## 4.1 EXPERIMENTS WITH NEURAL POLICIES AS PROXY HUMAN POLICIES

**Setup** To obtain neural experts with different performance levels as proxy human policies, we use RLPD (Ball et al., 2023) to train RL policies and load checkpoints at different training steps. We

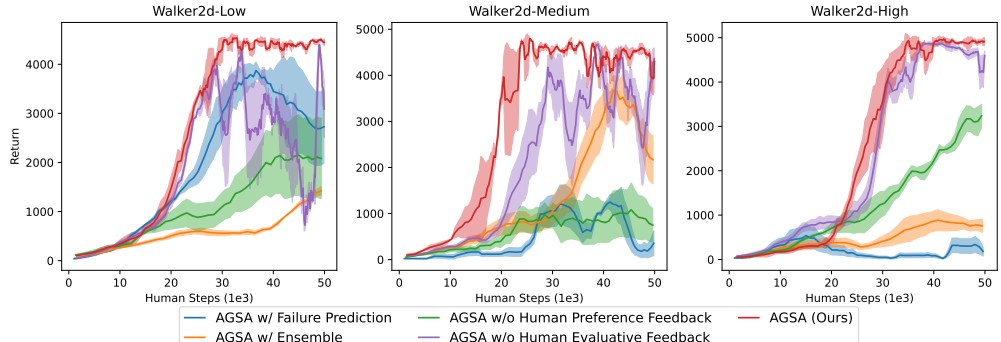

Figure 4: Learning curves of methods in ablation study. We consider the Walker2d environment. Full results are in Appendix C.5. The lines are average return across four different trials and the shadow areas denote the standard deviation.

Table 2: Results of ablation studies on different module combinations. The results are averaged and normalized in the same way as in Tab. 1.

| Domain | Expert Level | AGSA w/ Failure Prediction | AGSA w/ Ensemble | AGSA w/o Human Preference Feedback | AGSA w/o Human Evaluative Feedback | AGSA w/ $r_\psi$ as $r_\pi$ | AGSA (Ours) |
|---|---|---|---|---|---|---|---|
| Hopper | Low | **94.97** ± 3.46 | 88.29 ± 12.60 | 80.48 ± 14.35 | 83.28 ± 13.53 | 12.21 ± 20.47 | 94.18 ± 3.54 |
| | Medium | 91.79 ± 3.89 | 77.66 ± 13.93 | 80.03 ± 4.97 | **93.13** ± 1.71 | -4.64 ± 32.08 | 92.44 ± 3.83 |
| | High | 83.99 ± 5.08 | 77.47 ± 12.87 | 56.72 ± 2.70 | 70.86 ± 30.45 | -16.07 ± 25.62 | **95.79** ± 0.90 |
| | Average | 90.25 ± 4.14 | 81.14 ± 13.13 | 72.41 ± 7.34 | 82.42 ± 15.23 | -2.83 ± 26.06 | **94.14** ± 2.76 |
| Walker2d | Low | 58.90 ± 47.85 | 16.87 ± 5.66 | 38.14 ± 52.61 | 70.52 ± 37.57 | 103.50 ± 16.98 | **114.16** ± 2.17 |
| | Medium | -17.37 ± 17.04 | 40.84 ± 33.34 | -4.74 ± 23.45 | **111.51** ± 14.94 | 83.06 ± 31.73 | 109.38 ± 2.29 |
| | High | -23.06 ± 6.52 | -4.47 ± 9.30 | 75.26 ± 16.73 | 118.94 ± 6.74 | 101.37 ± 38.82 | **129.09** ± 2.98 |
| | Average | 6.16 ± 23.80 | 17.75 ± 16.10 | 36.22 ± 30.93 | 100.33 ± 19.75 | 95.98 ± 29.18 | **117.55** ± 2.48 |
| HalfCheetah | Low | 64.69 ± 4.61 | 8.43 ± 8.71 | 78.76 ± 0.55 | **84.31** ± 0.32 | 70.82 ± 3.86 | 83.01 ± 0.80 |
| | Medium | 82.38 ± 0.13 | 25.18 ± 13.70 | 85.69 ± 0.00 | 86.29 ± 0.82 | 85.17 ± 5.07 | **88.63** ± 0.20 |
| | High | 83.96 ± 1.34 | 32.75 ± 8.15 | 86.01 ± 0.92 | 85.36 ± 0.99 | **93.93** ± 0.74 | 89.66 ± 0.69 |
| | Average | 77.01 ± 2.03 | 22.12 ± 10.19 | 83.49 ± 0.49 | 85.32 ± 0.71 | 83.31 ± 3.22 | **87.10** ± 0.56 |

use policies at around 20%, 50%, and 100% performance levels compared with the optimal policy and term them as "low", "medium", and "high" policies, respectively. Detailed discussions on neural policies are in Appendix C.1. We follow previous preference-based RL methods (Lee et al., 2021; Xue et al., 2023b) and use comparisons of environment rewards to simulate human preferences feedback $p_t = P_\psi[\sigma \succ \sigma']$. Human evaluative feedback $I(s_t)$ is simulated by value function comparisons (Luo et al., 2024) with details discussed in Appendix C.1. For baseline algorithms, we mainly select previous agent-gated methods, including DAgger (Ross et al., 2011), EnsembleDAgger (Menda et al., 2019), Failure Detection (Liu et al., 2023a), and BCVA (Gokmen et al., 2023). RLIF (Luo et al., 2024) is also considered as the state-of-the-art human-gated algorithm in robotics locomotion. We use SAC (Haarnoja et al., 2018) to train both agents with $r_\psi$ and $r_\pi$. Detailed descriptions on the baseline algorithms are in Appendix C.2.

**Comparative Results** As shown in Tab. 1, our AGSA achieves the highest performance across all tasks and performance levels, demonstrating its ability of efficient learning from both optimal and imperfect simulated human policies. Imitation Learning-based methods, including DAgger, EnsembleDAgger, and failure detection, cannot outperform neural policies at each performance level and show suboptimal results. BCVA has poor performance in Hopper and Walker2d, due to the high variance in trajectory terminating signals. RLIF can outperform imperfect neural policies as value-gated intervention introduces additional information related to environment reward. But its performance drops significantly when value functions cannot provide accurate intervention signal, especially on HalfCheetah tasks.

**Ablation Study** The results of ablation studies on different module combinations are demonstrated in Tab. 2. We present performance curves for these methods in the Walker2d environment in Fig. 4. Full performance curves are left in Appendix C.5. The x-axis of Fig. 5 illustrates in the Walker2d environment, the experiments require 50k human intervention steps. AGSA takes an average of 30k human intervention steps to converge and is faster than all ablation methods. "AGSA w/ Failure Prediction" and "w/ Ensemble" are alternative approaches for constructing the gating agent. Both methods keep the learning agent training unchanged. "AGSA w/o Human Preference Feedback" and "w/o Human Evaluative Feedback" remove $r_\psi(s_t, a_t)$ and $I(s_t)$ respectively when computing the gating agent reward $r_G$. "AGSA w/ $r_\psi$ as $r_\pi$" refers to using the reward model $r_\psi$ trained from human preference feedback as the proxy reward $r_\pi$ to train the learning agent, in the same way as PbRL algorithms (Lee et al., 2021).

Table 3: Results of ablation studies on different values of hyperparameters $\lambda$ and $T$. The results are averaged and normalized in the same way as in Tab. 1.

| Domain | Expert Level | AGSA w/ $\lambda = 0.01$ | AGSA w/ $\lambda = 0.1$ | AGSA w/ $T = 2$ | AGSA w/ $T = 10$ | AGSA (Ours) w/ $\lambda = 0.03, T = 4$ |
|---|---|---|---|---|---|---|
| Hopper | Low | $85.73 \pm 8.12$ | $92.47 \pm 3.49$ | $84.70 \pm 0.50$ | $17.48 \pm 33.65$ | $\mathbf{94.18} \pm 3.54$ |
| | Medium | $84.24 \pm 14.62$ | $93.14 \pm 1.84$ | $\mathbf{96.43} \pm 0.83$ | $39.43 \pm 20.90$ | $92.44 \pm 3.83$ |
| | High | $85.18 \pm 12.62$ | $59.31 \pm 47.68$ | $53.22 \pm 8.74$ | $35.80 \pm 12.22$ | $\mathbf{95.79} \pm 0.90$ |
| | Average | $85.05 \pm 11.79$ | $81.64 \pm 17.67$ | $78.12 \pm 3.35$ | $30.91 \pm 22.26$ | $\mathbf{94.14} \pm 2.76$ |
| Walker2d | Low | $111.03 \pm 4.49$ | $90.48 \pm 36.48$ | $\mathbf{119.76} \pm 1.27$ | $55.92 \pm 62.15$ | $114.16 \pm 2.17$ |
| | Medium | $83.42 \pm 45.98$ | $102.30 \pm 20.67$ | $\mathbf{110.53} \pm 14.93$ | $85.65 \pm 32.98$ | $109.38 \pm 2.29$ |
| | High | $127.71 \pm 5.13$ | $\mathbf{130.58} \pm 2.33$ | $120.72 \pm 11.34$ | $50.05 \pm 22.41$ | $129.09 \pm 2.98$ |
| | Average | $107.39 \pm 18.53$ | $107.79 \pm 19.83$ | $117.00 \pm 9.18$ | $63.87 \pm 39.18$ | $\mathbf{117.55} \pm 2.48$ |
| HalfCheetah | Low | $83.33 \pm 0.82$ | $83.12 \pm 0.90$ | $\mathbf{86.86} \pm 0.56$ | $67.78 \pm 2.28$ | $83.01 \pm 0.80$ |
| | Medium | $87.99 \pm 0.36$ | $87.36 \pm 0.61$ | $\mathbf{89.09} \pm 0.24$ | $85.83 \pm 0.50$ | $88.63 \pm 0.20$ |
| | High | $88.66 \pm 0.23$ | $88.49 \pm 0.52$ | $88.72 \pm 1.22$ | $87.49 \pm 0.32$ | $\mathbf{89.66} \pm 0.69$ |
| | Average | $86.66 \pm 0.47$ | $86.32 \pm 0.68$ | $\mathbf{88.22} \pm 0.67$ | $80.37 \pm 1.03$ | $87.10 \pm 0.56$ |

Compared with the gating agent of AGSA that optimizes long-term performance, failure prediction and ensembled-based gating agent have comparable performance in the Hopper environment which is relatively simple to solve, but fail to achieve good performance in Walker2d and HalfCheetah environments. Compared with failure prediction and EnsembleDAgger in Tab. 1 that involve imitation learning, AGSA uses the proxy reward function $r_\pi$ to train the learning agent and obtains better performance in the Hopper and HalfCheetah environment. But $r_\pi$ is less effective when the gating agent is highly suboptimal, such as in the Walker2d environment with failure prediction.

According to the ablation results, AGSA also have degraded overall performance without either human preference feedback $p_t$ or human evaluative feedback $I(s_t)$, where preference feedback leads to larger performance gaps. As shown in Fig. 4, evaluative feedback is helpful to stabilize the training process with neural policies that have poorer performance. Meanwhile, the learning agent will not benefit from the preference reward model $r_\psi$ as the proxy reward $r_\pi$, as is employed in PbRL. This is because $r_\psi$ which is trained on human generated samples cannot accurately generalize to $(s, a)$ pairs that are more likely to be sampled by the learning agent. While $r_\psi$ is effective to train $Q_G$ with binary action space, such noisy reward signal may ruin the more complicated training of the learning agent.

We also conduct ablation studies on the hyperparameters in Tab. 3, including the preference reward ratio $\lambda$ and human intervention steps $T$. AGSA is robust with different scales of $\lambda$ and maintains superior performance compared with baseline algorithms. AGSA also fits well to fewer steps of continual intervention, but will have degraded performance if human demonstrations are extended to 10 steps. Large numbers of human control will increase distribution shift (Xu et al., 2022) of training samples and may lead to early termination due to imperfect interactions.

## 4.2 EXPERIMENTS WITH REAL HUMAN PARTICIPANTS

**Setup** We select the MetaDrive simulator (Li et al., 2023) to conduct experiments with real human participants that provide both low-level human demonstrations and high-level human feedback. Human participants are college students that are familiar with keyboard control but have little or no knowledge of the MetaDrive simulator. The instruction they receive is in Appendix C.3. Random control latency and environment speedup are inserted during training to simulate remote operation. Therefore, human participants are likely to provide imperfect interactions during training. For baseline algorithms, apart from EnsembleDAgger (Menda et al., 2019), Failure Detection (Liu et al., 2023a), BCVA (Gokmen et al., 2023), and RLIF (Luo et al., 2024) that are used in MuJoCo experiments, we consider imitation learning algorithms BC and GAIL (Ho & Ermon, 2016), as well as PVP (Peng et al., 2023) which is the state-of-the-art human gated algorithm. We use TD3 (Fujimoto et al., 2018) to train the agents.

For more accurate algorithm evaluation, we utilize the feature of procedure generation in MetaDrive and make a split of training and test environments with different maps and traffic. For the training process, we report the total human involvement steps that include steps of human monitoring and human taking actions in the simulator, total environment interaction steps of the learning agent and the human participants, and total safety cost which reflects the number of potential dangers exposed to the autonomous vehicle during training. We also report the episodic return, episodic safety cost of the learning agent, and the success rate as the test performance of the algorithms. The safety cost is a metric used to evaluate the safety performance of driving agents. It is incurred when the agent's

Table 4: Comparison of different human-in-the-loop methods in the MetaDrive environment. The human attention rate is given besides the human attention steps. We run all algorithms with three different seeds and report their average score as well as standard deviation.

| Method | Training | | | Testing | | |
|---|---|---|---|---|---|---|
| | Human Involvement Steps | Environment Interaction Steps | Total Safety Cost | Episodic Return | Episodic Safety Cost | Success Rate |
| BC | 30K (1.0) | - | - | 113.32 ± 10.21 | 2.17 ± 0.65 | 0.07 ± 0.02 |
| GAIL | 30K (0.015) | 2 M | 25.90 K ± 8.15 K | 81.51 ± 9.43 | **1.31** ± 0.23 | 0.0 ± 0.0 |
| EnsembleDAgger | 17.3K (0.865) | 20K | 55 ± 3.09 | 38.44 ± 3.98 | 8.38 ± 1.73 | 0.00 ± 0.00 |
| Failure Detection | 9.4K (0.47) | 20K | 66 ± 5.72 | 71.37 ± 15.24 | 1.92 ± 0.34 | 0.00 ± 0.00 |
| BCVA | 12.9K (0.645) | 20K | 74 ± 4.55 | 143.19 ± 12.28 | 5.04 ± 1.16 | 0.06 ± 0.01 |
| PVP | 20K (1.0) | 20K | 64 ± 2.05 | 174.71 ± 8.41 | 6.05 ± 0.85 | 0.17 ± 0.01 |
| RLIF | 20K (1.0) | 20K | 63 ± 1.25 | 169.54 ± 6.39 | 3.90 ± 1.22 | 0.19 ± 0.02 |
| AGSA (Ours) | 7.9K(0.395) | 20K | **51** ± 2.94 | **263.56** ± 8.22 | 5.78 ± 1.63 | **0.40** ± 0.02 |

vehicle collides with other objects or deviates from the designated road. This metric is crucial for assessing the agent's ability to navigate complex driving scenarios without accidents.

**Performance Comparison** Tab. 4 shows the performance comparison of the baseline algorithms and AGSA. AGSA requires the least human attention steps that is helpful for reducing human stress during training. Human-in-the-loop methods all have much lower training safety cost compared with the online imitation learning algorithm GAIL, with AGSA encountering the fewest safety violations. AGSA also obtains the highest test episodic return and test success rate, demonstrating its ability to train generalizable policies with imperfect human interactions. PVP and RLIF benefit from human monitoring and outperform agent-gated baseline algorithms. GAIL has the lowest test safety cost, mainly because of its poor performance and truncated trajectory.

**Survey on Human Participants** We design a user study to analyse the feelings of human participants during training. Detailed instructions are in Appendix C.3. We consider three metrics: *devotion* which is the degree of mental concentration, *anxiety* which measures the level of human stress and tension, and *performance* which is the human evaluation on agent behaviors. As shown in Tab. 5, AGSA exhibits more user-friendliness compared with baseline algorithms. The agent-gated framework frees human participants from constant monitoring and reduces the amount of human devotion to the experiment. It also leads to less human stress because humans are not directly responsible for safety violations and only in charge of providing feedback. AGSA also has the highest human rated performance level, in line with numerical evaluations.

Table 5: The result of user study. The maximum score is 5 for each metric. Metrics with (↑) are better with higher scores and vise versa.

| | PVP | RLIF | Failure Detection | AGSA |
|---|---|---|---|---|
| Devotion (↓) | 4.5 ± 0.5 | 4.7 ± 0.5 | 2.0 ± 0.9 | 1.6 ± 0.7 |
| Anxiety (↓) | 3.5 ± 1.0 | 4.3 ± 0.6 | 2.2 ± 0.7 | 2.0 ± 0.8 |
| Performance (↑) | 3.2 ± 0.8 | 2.2 ± 0.6 | 1.9 ± 0.7 | 4.5 ± 0.7 |

## 5 CONCLUSION

In this paper, we present a novel Agent-Gated Shared Autonomy (AGSA) framework for human-in-the-loop RL from imperfect human interactions, achieving reward-free, sample-efficient, and safe training of RL agents. Unlike previous approaches that rely on accurate human monitoring or optimal human demonstrations, we propose to learn from human evaluative and preference feedback. The gating agent is trained with both types of feedback to accurately model the long-term influence of control switch decisions. The learning agent is directly trained with the intervention decisions of the gating agent, mitigating the issue of suboptimal human demonstrations. We also provide both theoretical and empirical analysis to verify the effectiveness of AGSA.

**Limitations** AGSA only considers the interaction between one human participant, one learning agent and one environment. It will be interesting to scale AGSA up for interactions between $N$ human participants and $M$ learning agents, where $M \gg N$. In the MetaDrive experiment with real human participants, the action space has 2 dimensions, which makes human involvement relatively easy. Potential human interaction interfaces to solve tasks with higher-dimensional action spaces are discussed in in Appendix D.

**Ethics Statement** To conduct human-in-the-loop training, we pay human participants to invite them joining in the experiments. We ensure transparency by informing all participants about the aim of the experiments and how their interactions will be used. Every participant provide written consent, confirming they are fully aware and in agreement. As experiments are conducted in the simulator that supports pausing, human participants can pause or stop the experiment at any time, or temporarily refuse to intervene when the gating agent asks so. No human participants are injured and no real-world assets are effected by the experiments because all tasks are conducted through simulated environments. Experiments last no longer than thirty minutes and participants rest at least three hours after an experiment. During training and data processing, no personal information is collected in the trained models. The human-in-the-loop experiments are conducted under NTU-IRB-2024-1100.

**Acknowledgements** This research is supported by the National Research Foundation Singapore and DSO National Laboratories under the AI Singapore Programme (AISGAward No: AISG2-GC-2023-009), and NUS Start-up Grant A-0010106-00-00.

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

## A  ADDITIONAL RELATED WORK

**Reward-free RL**   To train RL policies without environment rewards, unsupervised skill discovery methods  (Eysenbach et al., 2019; Sharma et al., 2020) aim to maximize policy diversity and coverage.  Inverse RL methods learn the reward model by maximizing it on human-generated samples and minimizing it on agent-generated samples. But the it can be hard for learned reward models to generalize due to insufficient and suboptimal human demonstrations. Recently, preference-based RL (PbRL) methods (Lee et al., 2021; Kim et al., 2023) that learn the reward model from human preference pairs have achieved success in aligning large language models with human intentions (Ouyang et al., 2022; OpenAI, 2023; Touvron et al., 2023), primarily due to the relatively low cost of constructing large-scale human preference datasets. PbRL has also shown promising capabilities in continuous control tasks (Mattson et al., 2024; Hejna et al., 2024; Biyik et al., 2024a; Wilde et al., 2021; Biyik et al., 2024b). However, PbRL methods can be inefficient in training RL policies from scratch (Lee et al., 2021), as poor performing policies can hardly generate informative preference pairs. They may also encounter safety issues when policies trained from inaccurate reward models are interacting with the environment. Our AGSA method proposes to train a gating agent from human preferences that can guide and safeguard the learning agent.

## B  THEORY

### B.1  PROOFS

**Theorem B.1** (Restatement of Thm. 3.1). *With the gating policy $\pi_g$ defined in Eq. (1), the behavior policy $\pi_b$ defined in Eq. (*) has the following performance lower-bound:*

$$\eta(\pi_b) \geqslant \max\left\{\eta(\pi_h), \eta(\pi_l)\right\} - \frac{\varepsilon_r}{(1-\gamma)^2}, \tag{5}$$

*where $\varepsilon_r = \max_{s,a}|r(s,a) - r_\psi(s,a)|$ is the error of preference-based reward modelling.*

*Proof.*  We first show that by learning from preference-based reward $r_\psi$, the gating value function $Q_g$ has a bounded discrepancy with the value function $Q^{\pi_b}$ under the behavior policy $\pi_b$.

$$Q_g(s,1) = \mathbb{E}_{a\sim\pi_h(\cdot|s)}\left[r_\psi(s,a) + \gamma\mathbb{E}_{s'\sim T_l(\cdot|s,a),a'\sim\pi_g(\cdot|s')}[Q_g(s',a')]\right]$$
$$Q_g(s,0) = \mathbb{E}_{a\sim\pi_l(\cdot|s)}\left[r_\psi(s,a) + \gamma\mathbb{E}_{s'\sim T_l(\cdot|s,a),a'\sim\pi_g(\cdot|s')}[Q_g(s',a')]\right] \tag{6}$$

So we have

$$Q_g(s,1) - \mathbb{E}_{a\sim\pi_h(\cdot|s)}Q^{\pi_b}(s,a) = \mathbb{E}_{a\sim\pi_h(\cdot|s)}\left[r_\psi(s,a) - r(s,a)\right]$$
$$+ \gamma\mathbb{E}_{s'}\left[\mathbb{E}_{a'\sim\pi_g(\cdot|s')}Q_g(s',a') - \mathbb{E}_{a'\sim\pi_b(\cdot|s')}Q_b^\pi(s',a')\right], \tag{7}$$

$$Q_g(s,0) - \mathbb{E}_{a\sim\pi_l(\cdot|s)}Q^{\pi_b}(s,a) = \mathbb{E}_{a\sim\pi_l(\cdot|s)}\left[r_\psi(s,a) - r(s,a)\right]$$
$$+ \gamma\mathbb{E}_{s'}\left[\mathbb{E}_{a'\sim\pi_g(\cdot|s')}Q_g(s',a') - \mathbb{E}_{a'\sim\pi_b(\cdot|s')}Q_b^\pi(s',a')\right]. \tag{8}$$

$\mathbb{E}_{a\sim\pi_g(\cdot|s)}Q_g(s,a)$ can be computed by linearly combining Eq. (7) and Eq. (8):

$$\mathbb{E}_{a\sim\pi_g(\cdot|s)}Q_g(s,a) - \mathbb{E}_{a\sim\pi_b(\cdot|s)}Q^{\pi_b}(s,a)$$
$$= \mathbb{E}_{a\sim\pi_b(\cdot|s)}\left[r_\psi(s,a) - r(s,a)\right] + \gamma\mathbb{E}_{s'}\left[\mathbb{E}_{a'\sim\pi_g(\cdot|s')}Q_g(s',a') - \mathbb{E}_{a'\sim\pi_b(\cdot|s')}Q_b^\pi(s',a')\right]$$
$$= \mathbb{E}_{a\sim\pi_b(\cdot|s)}\left[r_\psi(s,a) - r(s,a)\right] + \gamma\mathbb{E}_{a\sim\pi_b(\cdot|s')}\left[r_\psi(s',a) - r(s',a)\right]$$
$$+ \gamma^2\mathbb{E}_{s''}\left[\mathbb{E}_{a'\sim\pi_g(\cdot|s'')}Q_g(s'',a') - \mathbb{E}_{a'\sim\pi_b(\cdot|s'')}Q_b^\pi(s'',a')\right]. \tag{9}$$

Iteratively computing the last term in Eq. (9), we have

$$\left|\mathbb{E}_{a\sim\pi_g(\cdot|s)}Q_g(s,a) - \mathbb{E}_{a\sim\pi_b(\cdot|s)}Q^{\pi_b}(s,a)\right| = \left|\mathbb{E}_{s'\sim d_s^{\pi_b}(\cdot),a\sim\pi_b(\cdot|s')}[r_\psi(s',a) - r(s',a)]\right|$$
$$\leqslant \frac{\varepsilon_r}{1-\gamma}. \tag{10}$$

Combining Eq. (10) with Eq. (7) and Eq. (8), we have

$$\left|Q_g(s,1) - \mathbb{E}_{a\sim\pi_h(\cdot|s)}Q^{\pi_b}(s,a)\right| \leqslant \varepsilon_r + \frac{\gamma\varepsilon_r}{1-\gamma} = \frac{\varepsilon_r}{1-\gamma},$$
$$\left|Q_g(s,0) - \mathbb{E}_{a\sim\pi_l(\cdot|s)}Q^{\pi_b}(s,a)\right| \leqslant \varepsilon_r + \frac{\gamma\varepsilon_r}{1-\gamma} = \frac{\varepsilon_r}{1-\gamma}. \tag{11}$$

When the gating action $a_g = 0$, we have $Q_g(s, 0) \geqslant Q_g(s, 1)$, so

$$\mathbb{E}_{a \sim \pi_h(\cdot|s)} Q^{\pi_b}(s, a) - \mathbb{E}_{a \sim \pi_l(\cdot|s)} Q^{\pi_b}(s, a) \leqslant Q_g(s, 1) - Q_g(s, 0) + \frac{2\varepsilon_r}{1 - \gamma} \leqslant \frac{2\varepsilon_r}{1 - \gamma} \tag{12}$$

for all state $s$. Similarly, when the gating action $a_g = 1$, we have $Q_g(s, 0) \leqslant Q_g(s, 1)$, so

$$\mathbb{E}_{a \sim \pi_l(\cdot|s)} Q^{\pi_b}(s, a) - \mathbb{E}_{a \sim \pi_h(\cdot|s)} Q^{\pi_b}(s, a) \leqslant \frac{2\varepsilon_r}{1 - \gamma}. \tag{13}$$

According to the performance difference lemma (Kakade & Langford, 2002), we have

$$
\begin{aligned}
\eta(\pi_h) - \eta(\pi_b) &= \frac{1}{1 - \gamma} \mathbb{E}_{s \sim d_{\pi_h}} \left[ \mathbb{E}_{a \sim \pi_h(\cdot|s)} A^{\pi_b}(s, a) \right] \\
&= \frac{1}{1 - \gamma} \mathbb{E}_{s \sim d_{\pi_h}} \left[ \mathbb{E}_{a \sim \pi_h(\cdot|s)} Q^{\pi_b}(s, a) - \mathbb{E}_{a \sim \pi_b(\cdot|s)} Q^{\pi_b}(s, a) \right] \\
&= \frac{1}{1 - \gamma} \mathbb{E}_{s \sim d_{\pi_h}} \left[ \mathbb{E}_{a \sim \pi_h(\cdot|s)} Q^{\pi_b}(s, a) - \pi_g(s) \mathbb{E}_{a \sim \pi_h(\cdot|s)} Q^{\pi_b}(s, a) \right. \\
&\qquad\qquad \left. - (1 - \pi_g(s)) \mathbb{E}_{a \sim \pi_l(\cdot|s)} Q^{\pi_b}(s, a) \right] \\
&= \frac{1}{1 - \gamma} \mathbb{E}_{s \sim d_{\pi_h}} \left[ (1 - \pi_g(s)) \left[ \mathbb{E}_{a \sim \pi_h(\cdot|s)} Q^{\pi_b}(s, a) - \mathbb{E}_{a \sim \pi_l(\cdot|s)} Q^{\pi_b}(s, a) \right] \right] \\
&\leqslant \frac{2\varepsilon_r}{(1 - \gamma)^2} \mathbb{E}_{s \sim d_{\pi_h}} \left[ (1 - \pi_g(s)) \right] \\
&= \frac{2\varepsilon_r (1 - \beta)}{(1 - \gamma)^2} \\
&\leqslant \frac{2\varepsilon_r}{(1 - \gamma)^2}.
\end{aligned}
\tag{14}
$$

Rearranging terms, we have

$$\eta(\pi_b) \geqslant \eta(\pi_h) - \frac{2\varepsilon_r}{(1 - \gamma)^2}. \tag{15}$$

A similar bound can be derived from Eq. (13) as

$$\eta(\pi_b) \geqslant \eta(\pi_l) - \frac{2\varepsilon_r}{(1 - \gamma)^2}. \tag{16}$$

So we have

$$\eta(\pi_b) \geqslant \max \left\{ \eta(\pi_h), \eta(\pi_l) \right\} - \frac{2\varepsilon_r}{(1 - \gamma)^2}, \tag{17}$$

which concludes the proof. $\qquad\square$

**Theorem B.2** (Restatement of Thm. 3.2). *Let $\tilde{\pi}$ be the optimal policy trained with proxy rewards $r_\pi(s, a)$. $\tilde{\pi}$ has the following performance lower bound:*

$$\eta(\tilde{\pi}) \geqslant \eta(\pi_h) - \frac{4\varepsilon_r}{(1 - \gamma)^2}. \tag{18}$$

*Proof.* The following proof borrows the main idea from RLIF (Luo et al., 2024). Since we assign negative rewards for human intervention steps with $Q_g(s, 1) > Q_g(s, 0)$, in order to maximize the cumulative proxy rewards, $\tilde{\pi}$ should make $Q_g(s, 1) \leqslant Q_g(s, 0)$. According to Eq. (11), we have

$$\mathbb{E}_{a \sim \tilde{\pi}(\cdot|s)} Q^{\pi_b}(s, a) - \mathbb{E}_{a \sim \pi_b(\cdot|s)} Q^{\pi_b}(s, a) \geqslant Q_g(s, 0) - Q_g(s, 1) - \frac{2\varepsilon_r}{1 - \gamma} \geqslant \frac{2\varepsilon_r}{1 - \gamma} \tag{19}$$

According to the performance difference lemma, we have

$$
\begin{aligned}
\eta(\tilde{\pi}) - \eta(\pi_b) &= \frac{1}{1 - \gamma} \mathbb{E}_{s \sim d_{\tilde{\pi}}} \left[ \mathbb{E}_{a \sim \tilde{\pi}(\cdot|s)} A^{\pi_b}(s, a) \right] \\
&= \frac{1}{1 - \gamma} \mathbb{E}_{s \sim d_{\tilde{\pi}}} \left[ \mathbb{E}_{a \sim \tilde{\pi}(\cdot|s)} Q^{\pi_b}(s, a) - \mathbb{E}_{a \sim \pi_b(\cdot|s)} Q^{\pi_b}(s, a) \right] \\
&\geqslant \frac{2\varepsilon_r}{(1 - \gamma)^2}.
\end{aligned}
\tag{20}
$$

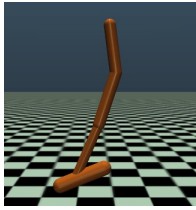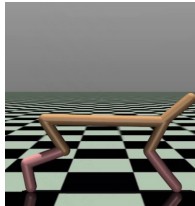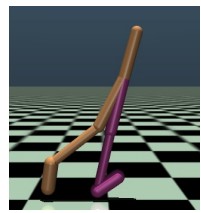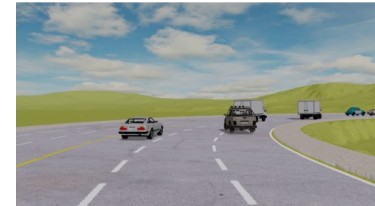

Figure 5: Environment visualizations of the robotics locomotion tasks Hopper, HalfCheetah, Walker2d, as well as the autonomous driving task.

| | Demonstrations | Preferences | Evaluations $I(s_t)$ | Value-based Intervention | Ground Truth Reward |
|---|---|---|---|---|---|
| DAgger and EnsembleDAgger | ✓ | × | × | × | × |
| Failure Prediction | ✓ | × | ✓ | ✓ | ✓ |
| BCVA | ✓ | × | × | × | ✓ (partially) |
| RLIF | ✓ | × | × | ✓ | ✓ |
| AGSA w/ Failure Prediction | ✓ | × | ✓ | ✓ | ✓ |
| AGSA w/ Ensemble | ✓ | × | × | × | × |
| AGSA w/o Preference Feedback | ✓ | × | ✓ | × | ✓ |
| AGSA w/o Evaluative Feedback | ✓ | ✓ | × | × | ✓ |
| AGSA w/ $r_\psi$ as $r_\pi$ | ✓ | ✓ | ✓ | × | ✓ |
| AGSA (Ours) | ✓ | ✓ | ✓ | × | ✓ |

Table 6: Comparison of environment information required by baseline and ablation methods.

Combining with Eq.( 14), we have

$$\eta(\tilde{\pi}) - \eta(\pi_h) = \eta(\tilde{\pi}) - \eta(\pi_b) + \eta(\pi_b) - \eta(\pi_h)$$
$$\geqslant \frac{2\varepsilon_r}{(1-\gamma)^2} + \frac{2\varepsilon_r}{(1-\gamma)^2} = \frac{4\varepsilon_r}{(1-\gamma)^2}, \tag{21}$$

which concludes the proof.

$\square$

### B.2 SAFETY BOUND

**Corollary B.3.** *With the gating policy $\pi_g$ defined in Eq. (1), the behavior policy $\pi_b$ defined in Eq. (*) has the following safety bound:*

$$C(\pi_b) \leqslant \min\{C(\pi_h), C(\pi_l)\} + \frac{\varepsilon_r}{(1-\gamma)^2}, \tag{22}$$

*where $\varepsilon_r = \max_{s,a}|r_c(s,a) - r_\psi(s,a)|$ is the error of preference-based reward modelling, $r_c(s,a)$ is the cost function, and $C(\pi) = \mathbb{E}_{\tau \sim d_0, \pi, T}\left[\sum_0^\infty \gamma^t r_c(s_t, a_t)\right]$ is the expected total cost of a trajectory.*

*Proof.* The proof can be obtained by replacing the $r(s_t, a_t)$ in the proof of Thm. 3.1 with $r_c(s_t, a_t)$. $\square$

## C ADDITIONAL EXPERIMENT DETAILS

### C.1 SETUP

The training tasks are visualized in Fig. 5. Neural policies used in the Hopper and Walker2d environment are the same as those in RLIF (Luo et al., 2024) experiments. For the Hopper environment, neural policies have about 20%, 70%, and 110% performance level compared with the optimal policy in D4RL (Fu et al., 2020). For the Walker2d environment, neural policies have about 15%, 40%, and 110% performance level compared with the optimal policy in D4RL (Fu et al., 2020). For the HalfCheetah environment, we train with RLPD (Ball et al., 2023) and use policies trained at 20k, 40k, 60k steps as neural policies with "low", "medium", and "high" policies. They have about 40%, 60%, and 100% relative performance, respectively. In MuJoCo experiments, we follow the approach in

RLIF (Luo et al., 2024) and obtain the simulated human intervention decision $\pi_g^{\text{human}}$ from the value function of the expert policy:

$$\pi_g^{\text{human}} = \begin{cases} 1, & \text{if } Q^\pi\left(s, \pi^{\text{human}}\left(s\right)\right) > Q^\pi\left(s, \pi_l(s)\right) \\ 0, & \text{otherwise.} \end{cases}$$

$I\left(s_t\right)$ is then obtained by comparing the gating agent's decision $\pi_g$ with $\pi_g^{\text{human}}$ when $\pi_g = 1$. If the gating agent proposes human intervention but human finds it unnecssary, i.e., $\pi_g^{\text{human}} = 0$, $I\left(s_t\right)$ will be set to 0, indicating a bad gating action. Different from RLIF that queries $\pi_g^{\text{human}}$ on each timestep, we only query $\pi_g^{\text{human}}$ for evaluative feedback when $\pi_g = 1$. We will add this discussion in the revision.

## C.2 BASELINES

We consider the following methods as baselines:

- BC: Use supervised learning to train the learning agent with the human-generated dataset.

- GAIL (Ho & Ermon, 2016): Use trajectory matching to train the learning agent. The learning agent needs full control to interact with the environment.

- DAgger (Ross et al., 2011): No gating agent is involved, with random control switches between the learning agent and the neural policy. The learning agent is trained by imitating the neural policy.

- EnsembleDAgger (Menda et al., 2019): The gating agent uses the output variance of the ensembled learning policy to determine when to let neural policies intervene.

- Failure Detection (Liu et al., 2023a): The gating agent is trained by imitating human gating behaviors. We follow previous approaches (Luo et al., 2024; Xue et al., 2023d) and use value function comparisons as a proxy of human gating. The learning agent is trained with imitation loss and next state reconstruction loss.

- BCVA (Gokmen et al., 2023): Use goal reaching rewards to learn proxy value functions. In robotics locomotion tasks, we set goal reaching rewards to -1 if the trajectory terminates. For the HalfCheetah environment without termination, we use the reward of the last step as the goal reaching reward. In the autonomous driving task, we send the goal reaching reward when the agent reaches the last checkpoint of the trajectory.

- RLIF (Luo et al., 2024): In robotics locomotion tasks with simulated human interactions, the learning agent is trained with human intervention signals that are generated by comparing the environment state-action value function $Q_{\text{env}}(s, a)$ between actions from the neural policy and the learning agent. As $Q_{\text{env}}$ itself is learned from a fixed replay buffer, such value-gated intervention can be inaccurate, which simulates the imperfect human intervention. In the autonomous driving tasks, we rely on human monitoring for human-gated training.

- PVP (Peng et al., 2023): Use human-gated training and directly optimizes the Q-value function of the learning agent to be close to +1 on human generated samples and close to -1 on agent generated samples.

In Tab. 6, we make a detailed comparison on which information is required by baselines and ablation methods in Tab. 1 and Tab. 2. All involved methods require expert demonstrations. Only DAgger, EnsembleDAgger, and AGSA with Ensemble (one of the ablation methods) do not require access to the ground truth reward, all of which have poor performances according to the experiment results.

## C.3 HUMAN STUDY

In human-gated training, human participants are instructed to perform active intervention whenever they identify that the learning agent is in dangerous or under-explored regions. The order of experiments with different approaches is randomized for each human participant. We use the following questionnaire to conduct user studies. Among the three metrics, "Performance" was chosen as a straightforward indicator of how effectively the agent completed tasks with minimal safety violations. "Anxiety" was included to evaluate the level of stress and fatigue experienced by participants. This measure captures the emotional and psychological responses to the agent's oscillations, unexpected behaviors, or limitations in intervention timing. It allows us to capture a

Table 7: Hyperparameters for the training algorithms.

| Algorithm | Hyperparameter | Values |
|---|---|---|
| Common | Batch Size | 256 |
| | Learning Rate | 3e-4 |
| | Weight Decay | 1e-3 |
| | Discount Factor $\gamma$ | 0.99 |
| | Hidden Dims | (256,256) |
| | $\tau$ for Target Network Update | 0.005 |
| DAgger | Pretrain Steps | 60,000 |
| | Steps Per Iteraction | 2500 |

| Algorithm | Hyperparameter | Values |
|---|---|---|
| EnsembleDAgger | Uncertainty Threshold | 0.03 (Hopper) |
| | | 0.1 (Walker2d) |
| | | 0.05 (HalfCheetah) |
| | | 0.01 (MetaDrive) |
| AGSA | Reward Balancing Ratio $\lambda$ | 0.03 |
| | Human Intervention Steps $T$ | 4 |

range of participant experiences and their immediate reactions to agent behavior. This stress-related metric offers insights beyond raw task performance, highlighting how smoothly the participants feel they can monitor the agent without stress from unpredictable events. "Devotion" was intended to measure the concentration required from participants, indicating whether the agent demanded continuous attention or if participants could rely on the agent to function independently. This feedback helps assess the cognitive workload AGSA imposes, reflecting whether participants feel the need to remain vigilant throughout training, thus directly correlating with workload.

---

**Performance**: Do you think the agent performs well with little safety violations when solving the task? The higher score the better.
Choices: 1, 2, 3, 4, 5

**Anxiety**: Do you think training with this agent is stressed? The higher score the more fatigue and stress. A lower score means you are more relaxed. Anxiety might come from many sources: Oscillating trajectory, unexpected behaviors, being unable to intervene on time, etc.
Choices: 1, 2, 3, 4, 5

**Devotion**: Do you think you have to keep focused when training with this agent? The higher score the more concentrated. A lower score means you do not need to take special care of the training agent.
Choices: 1, 2, 3, 4, 5

---

### C.4 HYPERPARAMETERS

We present the hyperparameters of the training algorithms in Tab. 7. "Common" refers to common hyperparameter settings shared by all algorithms. In EnsembleDAgger (Menda et al., 2019), human intervention will be triggered if the variance in proposed actions exceeds the uncertainty threshold. The thresholds need to be tuned in different environments. $\lambda$ and $T$ in AGSA keeps the same across all environments.

### C.5 RESULTS

We present the full learning curves of ablation studies in Fig. 6. In the Hopper environment, alternative gating algorithms facilitate more efficient training and achieve comparable overall performance. In the HalfCheetah environment, the "AGSA w/ Failure Prediction" method is also more efficient at the early stage of training. This is because alternative methods are either free of training or trained with more stable imitation loss, more quickly obtaining gating agents that have relatively good performance. Their performances also demonstrate the effectiveness of the proxy reward $r_\pi$ to train the learning agent from diverse gating agents. But in the Walker2d environment where these alternative methods have respective drawbacks, the performance of the learning agent will be degraded. Alternative methods cannot achieve higher asymptotic performance than AGSA either. We also illustrate the probability of AGSA requesting human intervention along the trajectory in the motivating example in Fig. 7. Although AGSA still assigns a little higher intervention probability on $t_4$ than on $t_1$, it successfully detects the potential danger in $t_2$ and $t_3$ and correctly assigns high intervention probabilities.

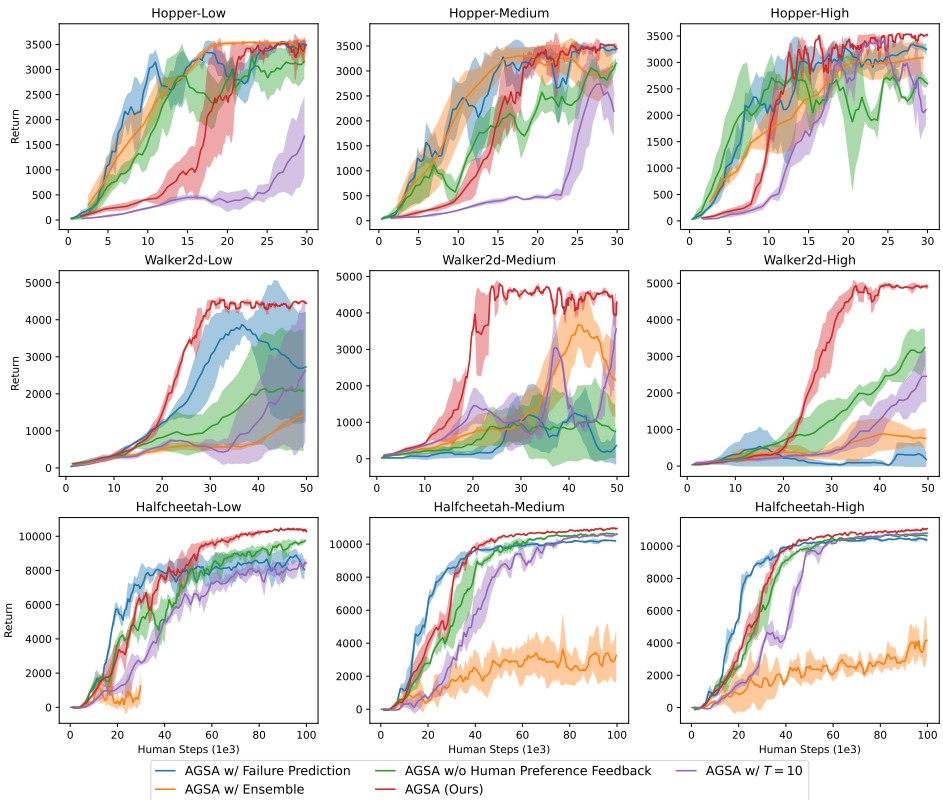

Figure 6: Learning curves of methods in ablation study. The lines are average return across four different trials and the shadow areas denote the standard deviation.

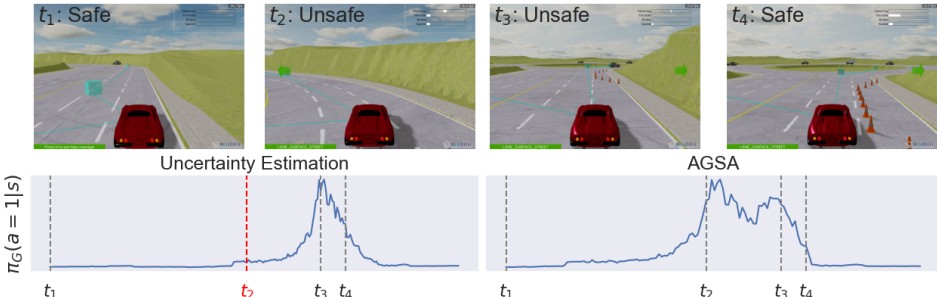

Figure 7: Probabilities of the gating agent requesting human intervention, with the uncertainty estimation method and AGSA.

# D    FUTURE WORK

We briefly discussed the limitation of AGSA in high-dimensional action spaces in the main paper. Such limitation is in fact a longstanding challenge in human-in-the-loop learning. Existing researches largely focus on tasks with small action spaces, such as simple robotic arms and autonomous driving. Therefore, we believe that generalization to higher-dimensional action spaces is orthogorical to the scope of this paper. Our method focuses on learning from imperfect human interaction and may still be helpful when human-in-the-loop RL is applied to higher-dimensional action spaces.

Nevertheless, we share our thoughts on potential challenges and solutions of this generalization. One of the major challenge of high-dimensional action spaces is that human participants may not be able to handle all dimensions simultaneously. Learning policies may also need an exponential increase in human interaction data to maintain the same demonstration coverage. This requires us to design

more scalable and efficient training algorithms. One potential solution is the hierarchical structure. A high-level planner agent with a low-dimensional action space can be designed to guide the behavior of the low-level controller agent with a high-dimensional action space. Human participants may only intervene the planner agent and provide demonstrations. Another potential solution is Policy Dissection (Li et al., 2022a), which is a simple yet effective frequency-based approach that aligns the intermediate representation of the learned neural controller with the kinematic attributes of the agent behavior. It has the potential of becoming a novel human interaction interface for tasks with high-dimensional action spaces.

