# OpenReview forum: "Policy Optimization under Imperfect Human Interactions with Agent-Gated Shared Autonomy"
_ICLR.cc/2025/Conference — ICLR 2025 Poster_

### Official Review · Reviewer_1nyz · 2024-10-31

**Soundness:** 3
**Presentation:** 3
**Contribution:** 3
**Rating:** 8
**Confidence:** 2

**Summary:**

The paper presents AGSA, a shared autonomy framework for learning from human feedback and gating when the robot passes control the human expert. The agent regards states that require intervention as undesirable, assigning negative rewards to state-action pairs that precede human intervention. AGSA trains a gating agent, and a learning agent. The gating agent requires human teachers to provide a binary signal on whether the current state is worth intervention, control for T steps when intervening, and provide a preference signal for whether the current segment is better than the previous segment. The learning agent is trained to optimize avoiding states that precede control switching to the human agent. The simulated results show that AGSA achieves high performance compared to baselines across multiple task and simulated performance levels. The ablation results indicate that the components are necessary for overall performance.

**Strengths:**

The manuscript is clear and the approach is well-described. I appreciate the motivating example, as it offers intuition about the limitations of ensemble-based uncertainty quantification and failure detection. While the experiments could use more detail about the quantity and type of data each method had access to, the results seem strong and indicate good performance of AGSA.

**Weaknesses:**

The paper presents a motivating example to demonstrate the limitations of ensemble-based uncertainty quantification and failure detection. However, it lacks an evaluation of how effectively the AGSA gating agent addresses these specific failure cases. The framework assumes a high level of expertise and understanding from the human operator, including familiarity with the robot's policy and decision-making criteria for interventions. Subsequently, these challenges are not addressed by the user study, which does not examine statistical testing nor provide sufficient details.

**Questions:**

In Section 3.1 Motivating example, what is the performance of the AGSA gating agent when handling the failure cases of uncertainty estimation and failure detection? It would helpful to show how AGSA addresses the issues in the motivating example.

It would be helpful to discuss the potential challenges of the assumptions AGSA places on the human. The human participant needs to have a very good understanding of the task, the robot’s policy, and how to change the robot’s policy in order to provide the 3 steps AGSA requires. For instance, is it possible the user could intervene but not know if the state necessarily required intervention?

How much time do humans have time to examine the intervention quality? How long did this take? How is it that they don’t have to “make real-time decisions that can be influenced by tiredness, carelessness, or network latency” when they still need to teleoperate the agent for T steps during intervention? If this occurs offline, then it makes sense how the human’s decisions would be free from latency. It would be great to comment on practical considerations and suitable tasks for a framework like AGSA.

Can state-action pairs that precede control switching by more than 1 timestep warrant avoiding? Currently the learning agent learns to avoid the single previous state, did you also consider larger windows?

For the real-human participant experiments, what was the total number of participants, and what was their breakdown? Was the study IRB approved? No statistical testing is done on the human subjects.

**Details Of Ethics Concerns:**

The paper conducts a human-subjects study, asking participants questions regarding devotion (mental load), anxiety (stress), and performance. The paper has applied, but as of the paper submission, not received IRB approval, based on the ethics statement. No statistical testing is performed. The study should be conducted after IRB approval.

---

> ### Author Response · Authors · 2024-11-15
> **Author Rebuttal (Part I)**
>
> We thank the reviewer for constructive feedback. We are encouraged that the reviewer acknowledged the clear and well-described approach, the intuitive motivating example, and strong empirical analysis. We provide explanations and discussions on the reviewer's concerns as follows.
>
> **Q1: It lacks an evaluation of how effectively the AGSA gating agent addresses the motivating example.**
>
> A: We agree that evaluating AGSA in the motivating example will be helpful for verifying its effectiveness. As a quick demonstration, we plot the intervention probabilities of AGSA in Appendix C.5, Figure 7 of the revised paper. Currently we do not place it in the main text due to constraints of space, but will try to squeeze some space in future revisions. According to the figure, AGSA still assigns a little higher intervention probability on $t_4$ than on $t_1$, which are safe timesteps. This may be because states that are similar to state $s_4$ are dangerous. Nevertheless, AGSA successfully detects the potential danger in timesteps $t_2$ and $t_3$ and correctly assigns higher intervention probabilities than safe timesteps. This demonstrates the effectiveness of AGSA's gating agent compared with baseline methods, namely uncertainty estimation and failure detection.
>
> **Q2: Experiments could use more detail about the quantity and type of data each method had access to.**
>
> A: In the following table, we make a detailed comparison on which information is required by baselines and ablation methods in Table 1 and Table 2. All involved methods are provided with expert demonstrations at different performance levels. Only DAgger, EnsembleDAgger, and AGSA with Ensemble (one of the ablation methods) do not require access to the ground truth reward, all of which have poor performances according to the experiment results. We briefly introduced how baseline algorithms are implemented in Appendix C.2. We apologize for not mentioning it in the main paper and will add this discussion in the revision.
>
> || Demonstrations | Preferences  | Evaluations $I(s_t)$ | Value-based Intervention | Ground Truth Reward      |
> | ----------------- | ---------- | ------------ | -------------------- | ------------------------ | ------------------------ |
> | DAgger and EnsembleDAgger| $\checkmark$   | $\times$     | $\times$             | $\times$                 | $\times$                 |
> | Failure Prediction| $\checkmark$   | $\times$     | $\checkmark$         | $\checkmark$             | $\checkmark$             |
> | BCVA| $\checkmark$   | $\times$     | $\times$             | $\times$| $\checkmark$ (partially) |
> | RLIF| $\checkmark$   | $\times$     | $\times$             | $\checkmark$| $\checkmark$             |
> | AGSA w/ Failure   Prediction| $\checkmark$   | $\times$     | $\checkmark$         | $\checkmark$             | $\checkmark$       |
> | AGSA w/ Ensemble| $\checkmark$   | $\times$     | $\times$             | $\times$| $\times$                 |
> | AGSA w/o Human Preference Feedback| $\checkmark$   | $\times$| $\checkmark$         | $\times$                 | $\checkmark$             |
> | AGSA w/o Human Evaluative Feedback| $\checkmark$   | $\checkmark$ | $\times$| $\times$                 | $\checkmark$             |
> | AGSA w/ $r_\psi$ as $r_\pi$| $\checkmark$   | $\checkmark$ | $\checkmark$         | $\times$                 | $\checkmark$             |
> | Failure Prediction + Preference-based RL (New baseline) | $\checkmark$   | $\checkmark$ | $\checkmark$| $\times$| $\checkmark$|
> | AGSA (Ours)| $\checkmark$   | $\checkmark$ | $\checkmark$         | $\times$| $\checkmark$             |
>
> **Q3: With respect to assumptions AGSA places on humans, is it possible the user could intervene but not know if the state necessarily required intervention?**
>
> A: This question is related to a fundamental question: Will it be easier for humans to make relative judgements than to provide absolute optimal decisions? As discussed in Lines 75-80 and backed up by several citations (Helson, 1964; Kahneman & Tversky,
> 2013; Ouyang et al., 2022), we assume this to be true in this paper and turn to relative human feedback, rather than direct human demonstrations, to train the gating agent. Some concrete examples of this statement include robotic manipulation, autonomous driving, language and image generation, and so on, where humans need to have good evaluation ability before being good at these tasks. We agree with the reviewer that there may be counter-examples, such as a pianist can perform based on muscle memory but is hard to give evaluative feedback. In these tasks, we may directly learn from the optimal demonstrations of humans, but we believe it will be out of the scope of this paper.

---

> ### Author Response · Authors · 2024-11-15
> **Author Rebuttal (Part II)**
>
> **Q4: Human interaction time, online training pipeline and suitable tasks for AGSA.**
>
> A: When training AGSA with real human participants, if a gating agent asks for human intervention at state $s_t$, the MetaDrive simulator will be paused and wait for the response of human participants. So humans have **unlimited time** (and may take a break) to check the intervention quality and give evaluation feedback. In real-world applications where we cannot pause the system, some rule-based safety policies can be exerted before human intervention. Apart from the simulator pause, human interactions are fully **online**. This includes continuous teleoperation for $T$ steps during intervention, which may be "influenced by tiredness, carelessness, or network latency". This is exactly what motivates us to take imperfections of human interactions into consideration and propose AGSA that can decently handle such imperfection.
>
> With respect to suitable tasks, AGSA is ideal for environments where human interventions are necessary but only occasionally required. Tasks like autonomous driving, industrial robot monitoring, or drone navigation involve long stretches of autonomous operation punctuated by critical intervention needs (e.g., avoiding obstacles, complex maneuvers). Meanwhile, AGSA is versatile enough to handle feedback from users with different skill levels. It could be applied to training simulations or professional development settings, such as driver training programs, to provide gradual autonomy based on user feedback.
>
> **Q5: Can state-action pairs that precede control switching by more than 1 timestep warrant avoiding?**
>
> A: This is a very good point. Monitoring for more than 1 timestep has the potential to make more accurate and robust intervention decisions. A direct application of this idea in AGSA is to request intervention if the gating value function $Q_g$ follows $Q_g(s,1)>Q_g(s,0)$ for continuous N timesteps.  We conduct preliminary studies and show the results in the following table.
>
> | Environment     | AGSA w/ N=2     | AGSA w/ N=3     | AGSA w/ N=1 (Ours) |
> | --------------- | --------------- | --------------- | ------------------ |
> | Walker2d-Low    | 114.29$\pm$2.44 | 110.53$\pm$0.05 | 114.16$\pm$2.17    |
> | Walker2d-Medium | 109.85$\pm$2.05 | 102.44$\pm$3.74 | 109.38$\pm$2.29    |
> | Walker2d-High   | 101.37$\pm$5.83 | 87.33$\pm$12.87 | 129.09$\pm$2.98    |
>
> According to the results, AGSA with 2 and 3 steps of continuous monitoring fail to achieve decent performance improvement. This may be because the continuous monitoring lowers the probability of human intervention. A further modification may be changing the intervention criteria to $Q_g(s,1)>Q_g(s,0)+\delta$, where $\delta$ is a hyperparameter. But this would also increase the complexity of AGSA and introduce a new hyperparameter. Given the preliminary results, we would prefer to keep the core AGSA algorithm as it is, add discussions on potential extensions in the revision, and leave them as future work.

---

> ### Author Response · Authors · 2024-11-15
> **Author Rebuttal (Part III)**
>
> **Q6: Details of real-human participant experiments**
>
> A: Sorry for the vague phrasing with respect to IRB. The IRB was approved before human studies but cannot be posted due to the double-blind policy. We will share the information when the paper is made public. The performance comparison results in Table 4 involves three human participants. The user study in Table 5 requires less devotion and involves ten human participants. As described in Lines 471-472, Human participants are college students that are familiar with keyboard control but have little or no knowledge of the MetaDrive simulator.
>
> To add statistical tests, we report the F-value, which is the ratio of the variance between group means to the variance within the groups. A larger F-value generally indicates a greater degree of difference between the means relative to the variance within each evaluation metric, namely Performance, Devotion, and Anxiety. We also report the p-value, which tells us whether the observed differences in means are statistically significant. A commonly used threshold is 0.05. If the p-value is below 0.05, we conclude that there are significant differences among the metric means. We further determine which specific algorithms differ. Each line in the following table represents a comparison between two algorithms:
> - **meandiff**: The difference in mean scores between two algorithms.
> - **p-adj**: The adjusted p-value for the pairwise comparison. If this value is below 0.05, the difference between the two groups is statistically significant.
> - **reject**: A Boolean value indicating whether the null hypothesis (that the means are equal) is rejected. If `True`, it means there is a significant difference between the two algorithms' means for that metric.
>
> The results for statistical tests are as follows.
>
> Metric: Performance; F-value: 26.78; p-value: 0.0000
>
> |     group1     |     group2     | meandiff | p-adj  | reject |
> | :------------: | :------------: | :------: | :----: | :----: |
> |      AGSA      | EnsembleDAgger |   -2.6   |  0.0   |  True  |
> |      AGSA      |      PVP       |  -1.25   | 0.0023 |  True  |
> |      AGSA      |      RLIF      |   -2.3   |  0.0   |  True  |
> | EnsembleDAgger |      PVP       |   1.35   | 0.0009 |  True  |
> | EnsembleDAgger |      RLIF      |   0.3    | 0.7876 | False  |
> |      PVP       |      RLIF      |  -1.05   | 0.0123 |  True  |
>
> We observe that it is statistically significant that AGSA is rated to have better performance than baseline algorithms.
>
> Metric: Anxiety; F-value: 16.40; p-value: 0.0000
> |group1  |group2| meandiff | p-adj  | reject |
> | :------------: | :------------: | :------: | :----: | :----: |
> |      AGSA      | EnsembleDAgger |   0.2    | 0.9527 | False  |
> |      AGSA      |      PVP       |   1.5    | 0.002  |  True  |
> |      AGSA      |      RLIF      |   2.3    |  0.0   |  True  |
> | EnsembleDAgger |      PVP       |   1.3    | 0.0085 |  True  |
> | EnsembleDAgger |      RLIF      |   2.1    |  0.0   |  True  |
> |      PVP       |      RLIF      |   0.8    | 0.1735 | False  |
>
> AGSA and EnsembleDAgger follows the agent-gated intervention that frees human participants from constant monitoring. Human participants feel less anxious because they are not directly responsible for safety violations and only in charge of providing feedback. The advantage of AGSA and EnsembleDAgger over PVP and RLIF in reducing participants' anxiety shows statistical significance.
>
> Metric: Devotion; F-value: 56.05; p-value: 0.0000
>
> |     group1     |     group2     | meandiff | p-adj | reject |
> | :------------: | :------------: | :------: | :------: | :----: |
> |      AGSA      | EnsembleDAgger |   0.4 | 0.5679   | False  |
> |      AGSA      |      PVP       |    2.9 | 0.0     |  True  |
> |      AGSA      |      RLIF      |    3.1 | 0.0     |  True  |
> | EnsembleDAgger |      PVP       |    2.5 | 0.0     |  True  |
> | EnsembleDAgger |      RLIF      |    2.7 | 0.0     |  True  |
> |      PVP       |      RLIF      |   0.2 | 0.9146   | False  |
>
> The agent-gated framework frees human participants from constant monitoring and reduces the amount of human devotion to the experiment. Compared with AGSA, EnsembleDAgger requires more devotion in average due to the poor performance, but the difference is not statistically significant. Meanwhile, the advantage of AGSA and EnsembleDAgger over PVP and RLIF in reducing participants' devotion shows statistical significance.

---

> ### Comment · Reviewer_1nyz · 2024-11-27
>
> Thank you for your clarifications, I have updated my scores.

---

### Official Review · Reviewer_p1os · 2024-11-02

**Soundness:** 3
**Presentation:** 3
**Contribution:** 2
**Rating:** 6
**Confidence:** 3

**Summary:**

The paper introduces AGSA (Agent-Gated Shared Autonomy), a novel framework designed to optimize reinforcement learning (RL) policy training in environments with imperfect human interaction. AGSA addresses key issues in human-in-the-loop learning, including reward-free training, safe exploration, and the challenges associated with suboptimal human control. Unlike existing methods, which often rely on perfect human intervention, AGSA uses a gating agent that determines when to involve human intervention, thus reducing the need for constant human oversight. This gating agent learns from human evaluative feedback and preferences regarding intervention timings and trajectories, aiming to minimize reliance on direct human monitoring or demonstrations. Theoretical insights provide performance bounds for the gating and learning agents, while experiments in continuous control environments with both simulated and real human participants highlight AGSA's advantages in safety, performance, and user-friendliness over previous approaches​.

**Strengths:**

- Innovative Framework for Human-in-the-Loop RL: AGSA introduces a unique approach to shared autonomy in reinforcement learning (RL) by using an agent-gated model that minimizes the need for perfect human intervention, enhancing training efficiency in reward-free settings.
- Rigorous Theoretical Proof: The paper offers theoretical guarantees, providing performance bounds for both the gating and learning agents, supported by clear proofs and analyses. Extensive experiments across diverse continuous control environments (e.g., robotic locomotion, autonomous driving) demonstrate AGSA’s advantages in training safety, efficiency, and robustness over other approaches.
- Potential Real-World Impact: AGSA’s design addresses the practical challenges of imperfect human feedback in RL, enabling more reliable and safe training in applications like autonomous driving and robotics. By ensuring safe exploration and learning efficiency even with suboptimal human intervention, AGSA broadens the scope of feasible real-world RL applications, especially in unpredictable, dynamic environments.
- Clear and Well-Written Presentation: The paper is well-organized, with detailed explanations of AGSA’s components, including gating agent training and feedback processing, making complex ideas accessible to readers.

**Weaknesses:**

- Reliance on Predefined Feedback Structures: AGSA depends on specific types of human feedback, namely evaluative and preference feedback, which may not always be feasible or scalable in real-world applications. A discussion on how to adapt AGSA for more passive or implicit feedback—where direct input from humans is minimal—could make the framework more versatile and user-friendly in broader applications.
- Generalization to Higher-Dimensional Action Spaces: The current evaluations are performed on tasks with relatively low-dimensional action spaces, such as robotic locomotion and simplified driving tasks. While these are challenging environments, it remains unclear how AGSA would perform in domains with more complex action requirements (e.g., humanoid robotics or high-degree-of-freedom manipulators). Testing or discussing potential challenges and solutions in these more complex settings would strengthen the generalizability of the approach.
- Evaluation of Human Burden: While AGSA claims to reduce the human workload, the metrics used for the user study might not be the most relevant. In the paper, authors used Performance, Anxiety and Devotion. Performance is a good metric and easy to understand. However, anxiety and devotion are comparably abstract. From the explanation in Appendix C.3, the explanation for the questionnaire is very vague. There would be some better and quantitative metrics for the user study. (Minor issue (line 890): "Choces" should be "Choices").
- Absence of Alternative Feedback Models for Gating: AGSA’s gating mechanism relies on a binary, evaluative feedback model that may not capture the nuances of continuous human feedback. Incorporating or discussing alternative feedback types, such as graded or probabilistic feedback, could further enhance the framework's adaptability to real-world conditions where binary feedback may be insufficient.

**Questions:**

See above sections.

---

> ### Author Response · Authors · 2024-11-15
> **Author Rebuttal (Part I)**
>
> We thank the reviewer for the constructive feedback. We are encouraged that the reviewer acknowledged the novel framework, rigorous theoretical proof, potential real-world impact, and well-written presentation of our paper. We provide explanations and discussions on the reviewer's four major concerns as follows.
>
> **Q1: Feasibility and scalability of evaluative and preference feedback**
>
> A: First, we would like to emphasize that these two types of feedback have already gained widespread usage in human-in-the-loop RL, preference-based RL, and finetuning large language models with RL from human feedback. There have also been large-scale human preference datasets that trains RL policies with billions of parameters. Nevertheless, we agree that there may be certain scenarios where only passive or implicit feedback is available. For example, in some mobile apps or web applications, users will only provide absolute evaluations on whether they enjoy the application or not. In such cases, we may manually create preference pairs for trajectories with different ratings from similar users and train the gating agent.
>
> **Q2: Generalization to Higher-Dimensional Action Spaces**
>
> A: We briefly discussed the limitation of AGSA in high-dimensional action spaces in lines 537-539. Such limitation is in fact a longstanding challenge in human-in-the-loop learning. Existing researches largely focus on tasks with small action spaces, such as simple robotic arms and autonomous driving. Therefore, we believe that generalization to higher-dimensional action spaces is orthogorical to the scope of this paper. Our method focuses on learning from imperfect human interaction and may still be helpful when human-in-the-loop RL is applied to higher-dimensional action spaces.
>
> Nevertheless, we share our thoughts on potential challenges and solutions of this generalization. They will also be added to the revision of our paper.
> - **Challenges:** One of the major challenge of high-dimensional action spaces is that human participants may not be able to handle all dimensions simultaneously. Learning policies may also need an exponential increase in human interaction data to maintain the same demonstration coverage. This requires us to design more scalable and efficient training algorithms.
> - **Potential solution 1: Hierarchical structure**   A high-level planner agent with a low-dimensional action space can be designed to guide the behavior of the low-level controller agent with a high-dimensional action space. Human participants may only intervene the planner agent and provide demonstrations.
> - **Potential solution 2: Policy dissection**   Policy dissection [1] is a simple yet effective frequency-based approach that aligns the intermediate representation of the learned neural controller with the kinematic attributes of the agent behavior. It has the potential of becoming a novel human interaction interface for tasks with high-dimensional action spaces.
>
> Q3: **The explanation for the questionnaire is very vague.**
>
> A: In the study, we selected **Performance**, **Anxiety**, and **Devotion** as metrics to assess both the technical and user-centric outcomes of AGSA.
> 1. **Performance** was chosen as a straightforward indicator of how effectively the agent completed tasks with minimal safety violations, as recognized in the comment.
> 2. **Anxiety** was included to evaluate the level of stress and fatigue experienced by participants. This measure captures the emotional and psychological responses to the agent’s oscillations, unexpected behaviors, or limitations in intervention timing. It allows us to capture a range of participant experiences and their immediate reactions to agent behavior. This stress-related metric offers insights beyond raw task performance, highlighting how smoothly the participants feel they can monitor the agent without stress from unpredictable events.
> 3. **Devotion** was intended to measure the concentration required from participants, indicating whether the agent demanded continuous attention or if participants could rely on the agent to function independently. This feedback helps assess the cognitive workload AGSA imposes, reflecting whether participants feel the need to remain vigilant throughout training, thus directly correlating with workload.
>
> We will add this discussion in the revision. In future work, we could integrate quantitative workload assessments such as measuring the participant heart rates as physiological indicators of stress. But this will require additional approvals and we may not be able to carry out during rebuttal period.

---

> ### Author Response · Authors · 2024-11-15
> **Author Rebuttal (Part II)**
>
> Q4: **Alternative Feedback Models for Gating**
>
> A: This is a very good point. Alternative evaluation feedback models, such as graded or probabilistic feedback, may learn the relative urgency or effectiveness of interventions, making it more sensitive to degrees of risk and control quality. Nevertheless, introducing alternative feedback types can increase the burden on human participants and the complexity of the learning algorithm. It will therefore be challenging to balance the efficiency and simplicity of the training algorithm. We would also like to mention that apart from binary evaluative feedback, AGSA also makes use of the preference feedback from human participants. Such kind of feedback model has been shown to be extremely scalable, with which large language models can be finetuned and aligned with human intentions. With the ability to learn from preference feedback, AGSA have the potential to handle more complex real-world tasks.
>
> **References**
>
> [1] Human-AI Shared Control via Policy Dissection. NeurIPS 2022.

---

> ### Comment · Reviewer_p1os · 2024-11-22
>
> Thanks for authors' detailed explanation. I'll keep my score.

---

> > ### Author Response · Authors · 2024-11-25
> > **Thanks for the Response**
> >
> > We thank the reviewer for their acknowledgement of our response.

---

### Official Review · Reviewer_o6YB · 2024-11-04

**Soundness:** 3
**Presentation:** 3
**Contribution:** 3
**Rating:** 6
**Confidence:** 4

**Summary:**

The paper presents an interactive imitation learning algorithm. The key idea is to learn a gating agent based on a ground truth intervention signals and a preference-based learned reward. The authors present experiments showing that their method achieves higher reward / success rates than comparison algorithms while being more convenient for human demonstrators.

**Strengths:**

Originality:
* The paper seems to be the first to use preferences in this context
* The reward formulation combining the ground truth intervention signal with the preference reward is interesting

Quality:
* The paper seems to achieve some impressive results
* The paper presents ablation results which verify the components of their method

Significance:
* Human in the loop IL is an important research direction and

**Weaknesses:**

Unfortunately there are several key weaknesses in the paper that make me think it needs some more work before it is ready for publication.
* As I mention below there are some missing details in the experiment section that make it difficult for me to understand the results. Most notably, I'm not sure how much data any of the algorithms were trained on or where the $I(s_t)$ values came from for AGSA.
* I'm concerned that the experiment in Table 1, which is pretty important to the paper, is not fair. As far as I can tell AGSA is the only algorithm which has access to the intervention signals $I(s_t)$ and the preferences (and by extension the ground truth reward)
* Some of the writing is confusing. Algorithm 1 helps a decent bit but I think there are a lot of missing details in the main text.

I'm concerned about the paper's fit for ICLR. As far as I can tell, the paper's key contributions are not so much about the learning method, since no part of their pipeline is very novel. Instead, my takeaway from this paper is that asking for preferences and GT intervention signals does not place much additional burden on supervisors and can help to improve HL learning performance. I think this paper is maybe a better fit for a robotics conference such as CoRL, ICRA or RSS than a learning conference like ICLR. Ultimately I think this issue is better decided by the AC so I am not factoring this into my score, but I feel it's an issue to raise.

**Questions:**

The following questions were ultimately answered in Algorithm 1, but until I got there I didn't find a clear answer in the main text of the paper:
* It says that humans provide a signal $I(s_t)$ which indicates whether $s_t$ is worthy of an intervention. Does this happen online or do humans go back and relabel the rollouts?
* When the humans interact with the environment (line 213) is that happening online based on some gating signal or is that a post-processing step as well?
* Line 230: what do 'current' and 'previous' segment mean? Is this temporal (ie {s_1, s_}, {s_3, s_4}) or current and previous versions of the policy?
* I don't see anywhere that it says how the gating agent is trained. Is it binary cross entropy on the GT gating values? Is it equivalent to the gating policy $\pi_g$?

Another quick question:
* In algorithm 1, are Q_G and Q_g the same thing?

Experiment Questions:
* How do you get the $I(s_t)$ values from the RL experts?
* What are the parameters of the experiment going into Table 1? How much data do all the policies get? What kind of reward do the expert policies get?
* How do the low and medium experts do so well for RLIF and AGSA?
* It seems to me that your method has much more information available to it than comparison methods (demos + preferences + GT $I(s_t)$ values. Are there any possible baselines that can use a similar amount of data, even possibly ablations on your method?
* I'm confused about what "safety cost" is in the human experiment. It says "the number of potential dangers exposed to the agent" but that's pretty vague

---

> ### Author Response · Authors · 2024-11-15
> **Author Rebuttal (Part I)**
>
> We thank the reviewer for in-depth and constructive reviews. We appreciate that the reviewer acknowledge the originality, quality, and significance of our paper. With respect to weaknesses and questions, we find the major concerns are 1. Fairness of comparative analysis with access to human evaluation and preference information; 2. Some missing details of the methods and experiments; 3. Whether the paper is fit for ICLR. We provide detailed discussions on these concerns as follows, and will add them in the revised version of our paper.
>
> **Q1: Fairness of results in Table 1 and baselines with a similar amount of information**
>
> A: With respect to the fairness of results in Table 1, we beg to differ in the statement
>
> >AGSA is the only algorithm which has access to $I(s_t)$, the preferences, and by extension the ground truth reward.
>
> In contrast, both the Failure Prediction (FP) and the RLIF algorithm have access to state-action value functions **trained with ground truth reward** to determine the human intervention timestep. With the human intervention signal, the evaluative feedback $I(s_t)$ can also be obtained by checking whether the gating agent makes the same intervention decision with the optimal one (See Q2 for details in obtaining $I(s_t)$). It is true that AGSA adopts a different approach of utilizing reward functions for simulating human feedback, but in the MuJoCo experiments it does not require additional information from the environment compared with FP and RLIF.
>
> In the following table, we make a detailed comparison on which information is required by baselines and ablation methods in Table 1 and Table 2. All involved methods require expert demonstrations. Only DAgger, EnsembleDAgger, and AGSA with Ensemble (one of the ablation methods) do not require access to the ground truth reward, all of which have poor performances according to the experiment results. We briefly introduced how baseline algorithms are implemented in Appendix C.2. We apologize for not mentioning it in the main paper and will add this discussion in the revision.
>
> | | Demonstrations | Preferences  | Evaluations $I(s_t)$ | Value-based Intervention | Ground Truth Reward      |
> | ---------------- | -------------- | ------------ | ------------ | --------- | ------------- |
> | DAgger and EnsembleDAgger| $\checkmark$   | $\times$     | $\times$| $\times$| $\times$|
> | Failure Prediction| $\checkmark$   | $\times$     | $\checkmark$         | $\checkmark$             | $\checkmark$             |
> | BCVA| $\checkmark$   | $\times$     | $\times$| $\times$| $\checkmark$ (partially) |
> | RLIF| $\checkmark$   | $\times$     | $\times$| $\checkmark$| $\checkmark$             |
> | AGSA w/ Failure   Prediction| $\checkmark$   | $\times$     | $\checkmark$         | $\checkmark$| $\checkmark$       |
> | AGSA w/ Ensemble| $\checkmark$   | $\times$     | $\times$| $\times$| $\times$|
> | AGSA w/o Human Preference Feedback| $\checkmark$   | $\times$     | $\checkmark$         | $\times$| $\checkmark$             |
> | AGSA w/o Human Evaluative Feedback| $\checkmark$   | $\checkmark$ | $\times$| $\times$| $\checkmark$             |
> | AGSA w/ $r_\psi$ as $r_\pi$| $\checkmark$   | $\checkmark$ | $\checkmark$         | $\times$| $\checkmark$             |
> | Failure Prediction + Preference-based RL (New baseline) | $\checkmark$   | $\checkmark$ | $\checkmark$         | $\times$                 | $\checkmark$|
> | AGSA (Ours)| $\checkmark$   | $\checkmark$ | $\checkmark$| $\times$| $\checkmark$|
>
> With respect to baselines with similar amount of information, we find it difficult to find one as AGSA is the first to introduce preference-based learning in human-in-the-loop training. Nevertheless, we have tried to include such comparison in the ablation studies, namely "AGSA with $r_\psi$ as $r_\pi$". Specifically, this method trains the learning agent with the learned preference-based reward model and trains the gating agent in the same way as AGSA, utilizing human preference and evaluation feedback. During rebuttal period, we tried another baseline algorithm with similar amount of information, as demonstrated in the table above (Failure Prediction + Preference-based RL). The new baseline algorithm trains the gating agent with failure prediction, i.e., supervised learning with simulated human intervention signals, and trains the learning agent with preference-based reward model. Performance comparison between these three algorithms with similar training information is listed as follows, where AGSA outperforms other two methods by a large margin.
>
> | Environment     | AGSA w/ $r_\psi$ as $r_\pi$ | Failure Prediction + Preference-based RL (New baseline) | AGSA (Ours)         |
> | --------------- | --------------- | --------------- | ------- |
> | Walker2d-Low    | 103.50$\pm$16.98| 52.48$\pm$37.81| **114.16**$\pm$2.17 |
> | Walker2d-Medium | 83.06$\pm$31.73| 70.75$\pm$25.13| **109.38**$\pm$2.29 |
> | Walker2d-High   | 101.37$\pm$38.82| 47.33$\pm$18.25| **129.09**$\pm$2.98 |

---

> ### Author Response · Authors · 2024-11-15
> **Author Rebuttal (Part II)**
>
> **Q2: Missing method and experiment details**
>
> A: Please check the bullet points for responses to each concerns.
> - **Data amounts:** The x-axis of Fig. 5 illustrates in the Walker2d environment, the experiments require 50k human intervention steps, where AGSA takes an average of 30k human intervention steps to converge. The total number of environment timesteps, which includes the shared autonomy of the learning agent and the expert policy, varies with different training algorithms due to different tendencies of resorting to human intervention. For example, in the Walker2d environment with low level of expert policy, the average human intervention rate is 0.21. So the total environment steps is 143k. This is notably fewer than standard off-policy RL algorithms such as TD3 and SAC, which typically require around 1M environment steps to converge. In Table 4 (MetaDrive results), the number of human involvement steps (human intervention+human monitoring) is listed in the second column. The human involvement rate is in the bracket alongside human involvement steps. AGSA needs fewer human involvement steps than baseline algorithms thanks to the agent-gated training pipeline without human monitoring.
> - **$I(s_t)$ in MuJoCo experiments:** In MuJoCo experiments, we follow the approach in RLIF [Luo et al., 2024] and obtain the simulated human intervention decision $\pi_g^{\text{human}}$ from the value function of the expert policy:
> $$
> \pi_g^{\text{human}}= \begin{cases}1, & \text { if } Q^{\pi}\left(s, \pi^{\text{human} }(s)\right)>Q^{\pi}(s, \pi_l(s)) \\\\ 0, & \text { otherwise. }\end{cases}
> $$
>    $I(s_t)$ is then obtained by comparing the gating agent's decision $\pi_g$ with $\pi_g^{\text{human}}$ when $\pi_g=1$.
>   If the gating agent proposes human intervention but human finds it unnecssary, i.e., $\pi_g^{\text{human}}=0$, $I(s_t)$ will be set to 0, indicating a bad gating action. Different from RLIF that queries $\pi_g^{\text{human}}$ on each timestep, we only query $\pi_g^{\text{human}}$ for evaluative feedback when $\pi_g=1$. We will add this discussion in the revision.
> - **$I(s_t)$ in MetaDrive experiments:** In MetaDrive experiments with human participants, when human intervention begins, we ask human participants to evaluate the intervention decision. This happens online with optional pauses, as discussed in the next bullet.
> - **Online human interaction:** None of the three steps of human interactions in AGSA (Sec. 3.2, lines 207-237) require post-processing or relabelling. A typical interaction pipeline is that when the gating agent asks for human intervention, the simulator will be paused and wait for the response of human participants. In real-world applications where we cannot pause the system, some rule-based safeguarding policies can be exerted before human intervention. When the human participants are available, they first give evaluative feedback. Then they give online demonstrations for a few steps, before providing preference feedback on these demonstrations.  This online interaction pipeline is in fact one of the advantages of AGSA over previous preference-based RL methods. It saves the burden for humans of reviewing previously sampled trajectories.
> - **Training the gating agent:** The "gating agent" is kind of a high-level term that summarize the autonomous gating process. In practice, the gating policy $\pi_g$ is in charge of generating human intervention decisions. Such decisions are made by querying the gating value function $Q_g$ (Eq. (1), lines 203-204). So in general we train $Q_g$ for a better gating agent. According to lines 250-251, the gating value function can be trained with standard value-based RL methods with the gating reward function $r_g$. We do not have ground truth gating values. According to lines 372-373, in MuJoCo experiments we use SAC to train the gating agent. According to lines 478, in MetaDrive experiments we use TD3 to train the gating agent.
> - **Expert policies:** We follow previous works (TS2C and RLIF) and train the expert policies with standard RL algorithms and ground truth environment rewards. We load the checkpoints of policies at around 20%, 50%, and 100% performance levels compared with the optimal policy and term them as “low”, “medium”, and “high” expert policies.
> - **Safety cost in human experiment:** Sorry for the vague definition. In the MetaDrive simulator, the **safety cost** is a metric used to evaluate the safety performance of driving agents. It is incurred when the agent's vehicle collides with other objects or deviates from the designated road. This metric is crucial for assessing the agent's ability to navigate complex driving scenarios without accidents. We will add this discussion in the revision.

---

> ### Author Response · Authors · 2024-11-15
> **Author Rebuttal (Part III)**
>
> **Q3: Paper's fit for ICLR**
>
> A: We believe this paper is a good fit for ICLR. Instead of focusing on the learning method, this paper is targeted at another important building block of ML and RL: the training data. In traditional RL, the training data is purely collected by the learning policy itself. Human-in-the-loop RL involve human participants to provide high-quality demonstration data. AGSA goes one step further, utilizing human feedback data to learn from imperfect human demonstration data. AGSA may also motivate future research on designing better learning methods for both human demonstration and human preference data. Meanwhile, there have been several papers on human-in-the-loop RL with little novelty in the learning method that get accepted in learning conferences like NeurIPS and ICLR: [1,2,3,4].
>
> Some other concerns are also addressed as follows.
>
> **Q4: In algorithm 1, are $Q_G$ and $Q_g$ the same thing?**
>
> A: Yes. Sorry for this inconsistency. Both of them denote the state-action value function of the gating agent. We will change $Q_G$ to $Q_g$ in the revision.
>
> **Q5: How do the low and medium experts do so well for RLIF and AGSA?**
>
> A: According to the table in Q1, RLIF assumes access to optimal intervention decisions that only let expert policy intervene when it has a higher state value than the learning agent. While the expert policy may be suboptimal, the value-based intervention mechanism helps to filter out imperfect expert actions. Instead, AGSA assumes access to accurate high-level feedback. If the expert performs poorly on some states, the gating agent will receive negative feedback for asking the expert to intervene. A well-performing gating agent trained with such feedback is able to detect imperfections in expert policies. The intervention decisions of the gating agent will therefore take the future performance of experts into consideration.
>
> **Q6: Current and previous segments**
>
> A: The "current" and "previous" segments refer to temporal orders in one trajectory, generated with the same policy. Take Fig. 1 (lines 55-62) as an example, when giving the preference signal at timestep $t_4$, we regard segment $t_1$ to $t_2$ as the "previous" segment, and $t_3$ to $t_4$ as the "current" segment.
>
> **Q7: Parameters of experiment in Table 1**
>
> A: The hyperparameters of the training algorithms are in Appendix C.4, Table 6. AGSA introduces two new hyperparameters, namely the reward balancing ratio $\lambda$ and the human intervention steps $T$. There effects are explored in the ablation studies in Table 3, where AGSA shows robustness to these hyperparameters.
>
> **References**
>
> [1] Efficient Learning of Safe Driving Policy via Human-AI Copilot Optimization. ICLR 2022.
>
> [2] Guarded Policy Optimization with Imperfect Online Demonstrations. ICLR 2023.
>
> [3] Learning from Active Human Involvement through Proxy Value Propagation. NeurIPS 2023.
>
> [4] RLIF: Interactive Imitation Learning as Reinforcement Learning. NeurIPS 2024.

---

> > ### Comment · Reviewer_o6YB · 2024-11-22
> >
> > Thanks to the authors for the detailed responses. The primary concern that led to my low score was the worry about unfair comparisons. The authors have gone to great lengths to address that concern, so I'll happily increase my score. I'm still worried about the novelty of the paper which is preventing me from raising it higher.
> >
> > Please make sure to update the final draft of the paper to address some of the readability concerns that I raised in my original review.

---

> > > ### Author Response · Authors · 2024-11-25
> > > **Thanks for the Response**
> > >
> > > We thank the reviewer for their acknowledgement of our response. The reviewer may kindly check the general response or the updated draft for discussions on the readability concerns.

---

### Official Review · Reviewer_ZseZ · 2024-11-04

**Soundness:** 4
**Presentation:** 4
**Contribution:** 3
**Rating:** 8
**Confidence:** 4

**Summary:**

The problem domain of this paper is human-in-the-loop learning where the human provides feedback to an AI agent by providing interventions. However, as opposed to human-gated interventions where the human decides when to intervene, this paper considers the setup where an external AI agent decides when the human should intervene. In addition, the paper attempts to address the problem of imperfect human feedback, i.e., the human demonstrations when they intervene may be suboptimal. To this end, the paper introduces AGSA, an Agent-Gated Shared Autonomy framework. Through experiments in robotics and autonomous driving tasks, AGSA shows improved training efficiency, safety, and user-friendliness compared to prior methods, even with variable-quality human input.

**Strengths:**

- The proposed method achieves more efficient and user-friendly learning from interventions compared to existing works.
- The paper provides both theoretical analyses and empirical evidence supporting the effectiveness of the proposed method.
- The proposed method enables learning from humans with varying level of expertise.
- The paper includes a human subject study.

**Weaknesses:**

- The proposed method requires the human to compare an intervention segment with the previous intervention segment. This brings the assumption that two segments are comparable. Since their initial states will be different, this is difficult to guarantee. This should be discussed.
- The paper says RLHF has not been thoroughly investigated in HL for continuous control tasks. This is not true. Even the original RLHF paper (Christiano et al. 2017) uses continuous control tasks. Similarly, I recommend the authors to check the works by Daniel S. Brown, Scott Niekum, Erdem Biyik, Nils Wilde, Dorsa Sadigh for further studies of RLHF and preference-based learning for continuous control tasks.
- The paper says "uncertainty cannot be aligned with human instructions." It is not clear what this means. Plus, there is no reference or study that supports this statement. Some clarification is needed.
- In line 141, "compares" should be "compare".
- The theoretical analyses presented in the paper are nice but the bounds do not seem useful in practice as they are very loose bounds. This should be discussed as a limitation.
- Line 290 has a broken reference to an equation.
- What is eta in Theorem 3.1. Is it properly defined in the paper?
- I understand the motivation behind using preference data. I agree it is a good way to mitigate the problems due to human suboptimality. But the imitation data could still be used in addition to preferences.
- The human subject studies should report the number of subjects and demographics information.
- Table 5 reports variances but I suggest adding statistical significance tests as well.
- Why is future tense used in line 547?
- The paper reports authors have applied for IRB. Is it not approved yet? It is not acceptable to conduct human subject studies before IRB approval.

**Questions:**

Please see the questions in the weaknesses section.

**Details Of Ethics Concerns:**

The paper states the team has applied for IRB, but it is not clear if it was already approved. If not, it is problematic that the paper already includes human subject studies.

**Post-Rebuttal Update: The authors clarified that their IRB was indeed approved. In this case, no ethics review is needed and I update my review accordingly.**

---

> ### Author Response · Authors · 2024-11-15
> **Author Rebuttal (Part I)**
>
> We thank the reviewer for the positive and constructive feedback. We are encouraged that the reviewer acknowledged the user-friendly design, theoretical soundness and empirical effectiveness of AGSA that can learn from humans with varying level of expertise. With respect to weaknesses and questions, we provide our responses as follows.
>
> **Q1: The assumption of comparable segments**
>
> A: When different segments are compared, their initial states will indeed be different. This may be different from traditional Preference-based RL methods. But in our setting, the compared segments are human intervention trajectories that will only be collected when the learning agent is in a potentially dangerous situation. Chances are that the gating agent is in early stages of training and do not request intervention in time. It can be hard for human participants to recover the agent to a safe place in such cases. Compared with other segments that interacts with the environment normally, it is easy to assign low preference on dangerous or even failed segments. When both segments are safe and operates smoothly, we also have an option to set the preference signal $p\_t=P\_\psi[\sigma \succ \sigma^{\prime}]$ to be 0.5, indicating the current segment $\sigma$ is equally preferred with the previous segment $\sigma'$.
>
> **Q2: Related work on RLHF and preference-based learning for continuous control tasks**
>
> A: Sorry for the vague statement. As discussed in Lines 34-37, human-in-the-loop learning (HL) refer to methods with humans actively intervene the training process. By saying "RLHF has not been thoroughly investigated in HL for continuous control tasks", we refer to the lack of utilization of human feedback in methods with human-agent shared autonomy. It is true that in some context HL may also include the practice of RLHF, where passive human feedback is also considered as "human-in-the-loop". We will make it clear in the revision. With respect to further studies of RLHF and PbRL, we thank the reviewer for pointing out the five leading authors in this area. After a thorough investigation, we find papers [1,2,3,4,5] to be related to our paper and will add discussions in the related work of the revised paper.
>
> **Q3: The sentence "uncertainty cannot be aligned with human instructions" is not clear**
>
> A: State uncertainty used in human-in-the-loop RL may not fully capture the nuanced, safety-critical cues that humans rely on when deciding to intervene. To be specific, the uncertainty-based gating policy $\pi_g^{\text{uncertainty}}$ is computed with
> $$
> \pi_g^{\text{uncertainty}}(s)= \begin{cases}1, & \text { if } \mathbb E_{a\sim\pi_l} ~\textbf{Var}[Q^{\pi_l}\left(s, a\right)]>\varepsilon \\\\ 0, & \text { otherwise, }\end{cases}
> $$
> where the variance of the value function is computed without direct human instructions. For example, when humans perceive that state $s_t$ requires intervention but $E_{a\sim\pi_l} ~\textbf{Var}[Q^{\pi_l}\left(s, a\right)]$ is low, there is no approach to adjusting the uncertainty-based intervention mechanism that directly aligns the human intention. Changing the hyperparameter $\varepsilon$ may be a indirect way, but it risks influencing all other intervention decisions.
>
> **Q4: The theoretical bounds do not seem useful**
>
> A: Thm 3.1 proposes a performance lower-bound that contains the higher performance among the human and learning policy. This demonstrates that the gating agent facilitates efficient exploration by leveraging human policies. But when human policies are suboptimal, the behavior policy will not be much worse than the learning policy itself, which deals with the issue of imperfect human demonstrations. Thm 3.2 mainly demonstrates the effectiveness of learning from the proxy reward function $r_\pi$, with a lower-bound containing the performance of the human policy. Although the error term $\varepsilon_r$ may not be accurately estimated, these two theorems provide high-level theoretical justifications on the effectiveness of AGSA. The theoretical analysis can also be compared with those in previous researches such as RLIF, where another assumption on perfect human intervention is required. We will add this discussion in the revision.
>
> **Q5: Broken reference in line 290**
>
> A: We think the reviewer might mean the Eq.(\*) in line 290. This is actually a reference to the equation with regard to the mixed behavior policy in lines 284-285: $\pi_b(\cdot|s)=(1-\pi_g(s))\pi_l(\cdot|s)+\pi_g(s)\pi_h(\cdot|s) (*)$.
>
> **Q6: What is $\eta$ in Thm 3.1?**
>
> A: $\eta$ is the expected return of a policy. In line 114 of the paper, we define $\eta\left(\pi_l\right)=\mathbb{E}_{\tau \sim d_0, \pi_l, T_l}\left[\sum_0^{\infty} \gamma^t r\left(s_t, a_t\right)\right]$.

---

> ### Author Response · Authors · 2024-11-15
> **Author Rebuttal (Part II)**
>
> **Q7: The imitation data could still be used in addition to preferences**
>
> A: We thank the reviewer for recognizing the benefits of preference data in handling human suboptimality. We agree with the reviewer that imitation data could indeed complement preference-based feedback, potentially enhancing AGSA’s learning efficiency and performance. For example, Imitation Learning can be especially valuable in the early stages of training, where the agent lacks a baseline policy and can benefit from imitation to acquire foundational skills quickly. Meanwhile, it can be a bit tricky and require additional adjustment to balance between the original AGSA losses and the imitation loss, which adds to the complexity of our method. In the revision, we will discuss the potential advantages and disadvantages of combining preference and imitation data in AGSA and outline these as potential extensions for further enhancing the framework.
>
> **Q8: Details of real-human participant experiments**
>
> A: Sorry for the vague phrasing with respect to IRB. The IRB was approved before human studies but cannot be posted due to the double-blind policy. We will share the information when the paper is made public. The performance comparison results in Table 4 involves three human participants. The user study in Table 5 requires less devotion and involves ten human participants. As described in Lines 471-472, Human participants are college students that are familiar with keyboard control but have little or no knowledge of the MetaDrive simulator.
>
> To add statistical tests, we report the F-value, which is the ratio of the variance between group means to the variance within the groups. A larger F-value generally indicates a greater degree of difference between the means relative to the variance within each evaluation metric, namely Performance, Devotion, and Anxiety. We also report the p-value, which tells us whether the observed differences in means are statistically significant. A commonly used threshold is 0.05. If the p-value is below 0.05, we conclude that there are significant differences among the metric means. We further determine which specific algorithms differ. Each line in the following table represents a comparison between two algorithms:
> - **meandiff**: The difference in mean scores between two algorithms.
> - **p-adj**: The adjusted p-value for the pairwise comparison. If this value is below 0.05, the difference between the two groups is statistically significant.
> - **reject**: A Boolean value indicating whether the null hypothesis (that the means are equal) is rejected. If `True`, it means there is a significant difference between the two algorithms' means for that metric.
>
> The results for statistical tests are as follows.
>
> Metric: Performance; F-value: 26.78; p-value: 0.0000
>
> |     group1     |     group2     | meandiff | p-adj  | reject |
> | :------: | :------: | :------: | :----: | :----: |
> |AGSA| EnsembleDAgger |   -2.6   |  0.0   |  True  |
> |AGSA|PVP|  -1.25   | 0.0023 |  True  |
> |AGSA|RLIF|   -2.3   |  0.0   |  True  |
> | EnsembleDAgger |PVP|   1.35   | 0.0009 |  True  |
> | EnsembleDAgger |RLIF|   0.3    | 0.7876 | False  |
> |PVP|      RLIF|  -1.05   | 0.0123 |  True  |
>
> We observe that it is statistically significant that AGSA is rated to have better performance than baseline algorithms.
>
> Metric: Anxiety; F-value: 16.40; p-value: 0.0000
> |group1  |group2| meandiff | p-adj  | reject |
> | :----: | :-----: | :------: | :----: | :----: |
> |AGSA| EnsembleDAgger |   0.2    | 0.9527 | False  |
> |AGSA|PVP|   1.5    | 0.002  |  True  |
> |AGSA|RLIF|   2.3    |  0.0   |  True  |
> | EnsembleDAgger |PVP|   1.3    | 0.0085 |  True  |
> | EnsembleDAgger |RLIF|   2.1    |  0.0   |  True  |
> |PVP|RLIF|0.8| 0.1735 | False  |
>
> AGSA and EnsembleDAgger follows the agent-gated intervention that frees human participants from constant monitoring. Human participants feel less anxious because they are not directly responsible for safety violations and only in charge of providing feedback. The advantage of AGSA and EnsembleDAgger over PVP and RLIF in reducing participants' anxiety shows statistical significance.
>
> Metric: Devotion; F-value: 56.05; p-value: 0.0000
>
> |     group1     |     group2     | meandiff | p-adj | reject |
> | :------: | :------------: | :------: | :------: | :----: |
> |AGSA| EnsembleDAgger |   0.4 | 0.5679   | False  |
> |AGSA|PVP|2.9 | 0.0|  True  |
> |AGSA|RLIF|3.1 | 0.0|  True  |
> | EnsembleDAgger |PVP|    2.5 | 0.0     |  True  |
> | EnsembleDAgger |RLIF|    2.7 | 0.0     |  True  |
> |PVP|RLIF|   0.2 | 0.9146   | False  |
>
> The agent-gated framework frees human participants from constant monitoring and reduces the amount of human devotion to the experiment. Compared with AGSA, EnsembleDAgger requires more devotion in average due to the poor performance, but the difference is not statistically significant. Meanwhile, the advantage of AGSA and EnsembleDAgger over PVP and RLIF in reducing participants' devotion shows statistical significance.

---

> ### Author Response · Authors · 2024-11-15
> **Author Rebuttal (Part III)**
>
> **References**
>
> [1] Representation Alignment from Human Feedback for Cross-Embodiment Reward Learning from Mixed-Quality Demonstrations. C Mattson, A Aribandi, DS Brown.
>
> [2] Contrastive prefence learning: Learning from human feedback without rl. J Hejna, R Rafailov, H Sikchi, C Finn, S Niekum, WB Knox, D Sadigh.
>
> [3] Batch Active Learning of Reward Functions from Human Preferences. E Biyik, N Anari, D Sadigh.
>
> [4] Learning Reward Functions from Scale Feedback. N Wilde, E Bıyık, D Sadigh, SL Smith.
>
> [5] Active preference-based Gaussian process regression for reward learning and optimization. E Bıyık, N Huynh, MJ Kochenderfer, D Sadigh.

---

> > ### Comment · Reviewer_ZseZ · 2024-11-24
> >
> > I thank the authors for their response. I already recommend the acceptance of the paper, so I will keep my score. I updated my comment about the "ethics review" based on the authors' clarification.

---

> > > ### Author Response · Authors · 2024-11-25
> > > **Thanks for the Response**
> > >
> > > We thank the reviewer for their positive feedback throughout the rebuttal and the acknowledgement on our ethics statement.

---

### Author Response · Authors · 2024-11-25
**Global feedback and rebuttal summarization**

We thank the reviewers for their time and greatly appreciate their constructive comments. We are particularly encouraged that reviewers acknowledged many strengths of our paper, including technical novelty, well-motivated approach, theoretical soundness, and empirical effectiveness. We prepared detailed rebuttal for reviewers' concerns on paper weaknesses. We are happy that Reviewers ZseZ, o6YB, and p1os acknowledged our responses, with Reviewer o6YB kindly raising the score. (Update: Now all four reviewers respond to the rebuttal. Thank you!) The paper has been revised according to the rebuttal discussions. Detailed revisions, which are marked red in the updated pdf file, include:
- In Section 3.4, we discuss the theoretical bounds in more details. (Reviewer ZseZ)
- In Section 4 and Appendix C.1, we add experiment details for the readability concerns of Reviewer o6YB.
- In the ethics statement, we make clear the IRB approval statement.
- In Appendix A, we add discussions on more related work. (Reviewer ZseZ)
- In Appendix C.2, we make a detailed comparison on which information is required by baselines and ablation methods. (Reviewer o6YB)
- In Appendix C.3, we make additional explanations for the questionnaire. (Reviewer p1os)
- In Appendix C.5, we plot the intervention probabilities of AGSA in the motivating example. (Reviewer 1nyz)
- In Appendix D, we discuss the challenges and possibilities to generalize AGSA to higher-dimensional action spaces (Reviewer p1os).

As the author-reviewer discussion period is ending, we would be happy to clarify any issues with the paper and remain open to further revisions. We thank the reviewers again for their time and for their thoughtful consideration of our submission.

---

### Meta-Review · Area_Chair_hZJr · 2024-12-19

**Metareview:**

This paper addresses the challenge of human-in-the-loop learning for autonomous policy generation and proposes an innovative agent-gated approach that effectively incorporates suboptimal human feedback.

The reviewers recognize the significance of the problem, acknowledge the novelty of the proposed method, and find the results compelling. The final ratings unanimously indicate that the paper merits acceptance.

One reviewer raised a concern regarding the topic's fit for ICLR. After a thorough review of the ICLR subject areas, I am confident that this paper aligns well with the conference's scope, particularly under the domains of general machine learning and its applications to robotics and autonomy.

**Additional Comments On Reviewer Discussion:**

The authors clarified questions regarding experimental details and addressed reviewers' comments on improving presentation of the work. Reviewers indicated satisfaction with the paper after the discussion.

---

### Decision · Program_Chairs · 2025-01-22

Accept (Poster)